# Kinetic and structural mechanism for DNA unwinding by a non-hexameric helicase

Sean P. Carney[1,8], Wen Ma [2,3,6,8], Kevin D. Whitley[2,3,7], Haifeng Jia[4], Timothy M. Lohman[4], Zaida Luthey-Schulten [1,3] & Yann R. Chemla [3,5✉]

UvrD, a model for non-hexameric Superfamily 1 helicases, utilizes ATP hydrolysis to translocate stepwise along single-stranded DNA and unwind the duplex. Previous estimates of its step size have been indirect, and a consensus on its stepping mechanism is lacking. To dissect the mechanism underlying DNA unwinding, we use optical tweezers to measure directly the stepping behavior of UvrD as it processes a DNA hairpin and show that UvrD exhibits a variable step size averaging ~3 base pairs. Analyzing stepping kinetics across ATP reveals the type and number of catalytic events that occur with different step sizes. These single-molecule data reveal a mechanism in which UvrD moves one base pair at a time but sequesters the nascent single strands, releasing them non-uniformly after a variable number of catalytic cycles. Molecular dynamics simulations point to a structural basis for this behavior, identifying the protein-DNA interactions responsible for strand sequestration. Based on structural and sequence alignment data, we propose that this stepping mechanism may be conserved among other non-hexameric helicases.

[1] Department of Chemistry, University of Illinois at Urbana-Champaign, Urbana, IL 61801, USA. [2] Center for Biophysics and Quantitative Biology, University of Illinois at Urbana-Champaign, Urbana, IL 61801, USA. [3] Center for the Physics of Living Cells, University of Illinois at Urbana-Champaign, Urbana, IL 61801, USA. [4] Department of Biochemistry and Molecular Biophysics, Washington University in St. Louis School of Medicine, St. Louis, MO 63110, USA. [5] Department of Physics, University of Illinois at Urbana-Champaign, Urbana, IL 61801, USA. [6]Present address: University of California, San Diego, CA, USA. [7]Present address: Newcastle University, Newcastle upon Tyne, UK. [8]These authors contributed equally: Sean P. Carney, Wen Ma. ✉email: ychemla@illinois.edu

Molecular motors couple chemical energy into directed motion used to carry out their many functions in the cell. How motors step with each round of catalysis presents an important clue for understanding their molecular mechanism. Together with analysis of the dwell times between successive steps and their statistical fluctuations, measurements of stepping behavior have provided critical insight into molecular motors' mechanisms of operation and coordination.

Helicases are a class of molecular motors that use nucleoside triphosphate (NTP) hydrolysis to translocate on single-stranded (ss) nucleic acids (NA) and unwind double-stranded (ds) NA[1,2]. They are involved in nearly all facets of nucleic acid metabolism in the cell, with roles in replication, recombination, DNA repair, transcription, translation, and splicing[1–4]. *E. coli* helicase UvrD couples ATP hydrolysis to 3′ to 5′ translocation on ssDNA and unwinding of dsDNA[5–7] and is a prototype for understanding the molecular mechanism of the non-hexameric Superfamily 1 (SF1) helicases, one of the two largest structural classes of helicases[4,8]. UvrD is involved in several aspects of bacterial genome maintenance, including methyl-directed mismatch repair[9,10], nucleotide excision repair[11], plasmid replication[12], replication fork reversal[13], transcription-coupled repair[14], and recombination[15,16]. It consists of four subdomains[17] (Fig. 1a), characteristic of SF1 helicases[8]: two RecA-like subdomains, 1A and 2A, which comprise the motor core that binds ssDNA and couples ATP hydrolysis to translocation on DNA[17,18], and the two accessory subdomains, 1B and 2B.

Prior structural, biochemical, and single-molecule studies have provided estimates for the step sizes of UvrD-like helicases, from which mechanisms of ssDNA translocation and duplex unwinding have been proposed. Crystal structures of UvrD and its homolog PcrA on a ss/dsDNA junction have been used to infer an unwinding step size of 1 base pair (bp) unwound per ATP hydrolyzed for UvrD and a translocation step size of 1 nucleotide (nt) per ATP for PcrA[17,19]. This value agrees with ensemble kinetic measurements of the chemical step size (or ATP coupling ratio)—the distance advanced for each ATP hydrolyzed[2,3]—of 1 nt/ATP during ssDNA translocation by both helicases[20,21]. These structural data have led to a proposed model of helicase translocation in which the motor subdomains 1A and 2A inchworm along ssDNA by 1 nt per ATP hydrolyzed, each leading to unwinding of the duplex by 1 bp[17,19].

However, single-turnover ensemble kinetic studies of UvrD determined a kinetic step size—the distance advanced between successive rate-limiting steps[2,3]—of 4–5 nt for ssDNA translocation at saturating ATP[20,22,23] (7 nt at 10 μM ATP[20]). Similarly, non-unitary kinetic step sizes for UvrD unwinding have been reported. In vitro, DNA unwinding requires a dimer of UvrD[2,22,24–27], although a monomer can be activated by the application of force[7,28] or the presence of accessory proteins[29,30]. Ensemble pre-steady state[6] and single-molecule FRET[25] studies obtained unwinding kinetic step sizes of 4–5 bp and ~3 bp for UvrD dimers, respectively. A noise analysis[31] of single-molecule magnetic tweezers data was used to estimate a kinetic step size of ~4 bp for UvrD dimers[24] and ~6 bp for monomers[7] under force. Ensemble fluorescence studies determined values in the range of ~3–4 bp for monomeric UvrD complexed with partner protein MutL[29]. Non-unitary kinetic step sizes ranging from 4 to 9 bp have been reported for a number of other SF1 helicases, including RecBCD[32], Dda[33], and TraI[34]. Kinetic estimates of step size suffer from averaging over undetected short-lived intermediates, molecular heterogeneity, and rare behaviors like pausing and backsliding. The development of high-resolution single-molecule methods has resulted in the most direct measurements of helicase step size to date. Several helicases have thus been found to unwind dsDNA in 1-bp steps, including SF2 helicase XPD[35] and SF1 helicase Pif1[36], and to translocate on ssDNA in 1-nt steps,

such as SF2 helicase Hel308[37]. However, non-unitary unwinding step sizes ranging between 2 and 11 bp have also been reported for SF2 helicases RecQ[36,38] and NS3[39–41].

Several mechanisms have been proposed to explain non-unitary step sizes. One model is that certain helicases can melt multiple base pairs simultaneously from hydrolysis of a single ATP[42,43]. Spring-loaded mechanisms posit that translocation and unwinding are temporally segregated; while the motor core subdomains inchworm along one strand 1 nt at a time, unwinding is delayed until a third subdomain springs forward to melt several base pairs in one burst[40,44]. The order of the temporally segregated unwinding and translocation is reversed in alternate mechanisms in which multiple base pairs are melted simultaneously in the absence of ATP, followed by rapid translocation on nascent ssDNA in 1 nt/ATP steps[45,46]. Pausing models propose that both ssDNA translocation and duplex unwinding occur in (rapid) 1-nt or -bp steps for each ATP hydrolyzed, but that a rate-limiting pause much longer than the dwell times between individual steps occurs after several such steps[23]. In delayed-release mechanisms, 1 bp of DNA is melted per ATP, but the unwound strands are sequestered and released only after multiple unwinding rounds[36,38,41]. While these models can explain the observation of non-unitary step sizes, and in the case of the latter four can help reconcile seemingly disparate step size estimates, a firm structural basis for these proposed stepping mechanisms has remained elusive.

Here, we report the direct measurement of stepping dynamics of UvrD processing a DNA hairpin using high-resolution optical tweezers, and we propose a stepping mechanism based on our single-molecule results and molecular dynamics (MD) simulations of UvrD on a DNA fork junction. While UvrD dimers are required to unwind DNA in vitro[2,22,24–27], we apply force on the DNA to activate UvrD monomer unwinding[7,28]. Monomeric UvrD was previously shown to translocate on both strands of a hairpin, leading to intermittent periods of duplex unwinding and re-zipping (Supplementary Fig. 1)[7,28]. We measure an average step size of ~3 bp for monomeric UvrD for both unwinding and re-zipping across all ATP concentrations and applied force. We also show that dimeric UvrD exhibits the same 3 bp average unwinding step size. However, a large variance in step size, evidence for subpopulations of step sizes smaller than 3 bp, and a periodicity of ~0.5 bp in the step size distribution show that the fundamental stepping unit is not 3 bp. Integrating our results and those from previous studies, we propose a mechanism consistent with delayed-release models in which UvrD translocates and unwinds 1 nt/bp at a time but sequesters both newly formed ssDNA strands, releasing them in a non-uniform manner after several rounds, the number of which is variable but typically 3. MD simulations of the UvrD-DNA complex support this model and suggest a structural basis for strand sequestration and release, identifying basic amino acid residues in the motor subdomains of UvrD that contact the released DNA strands and sequester them into loops. The distribution of loop sizes from these simulations recapitulates the experimental step size distribution. Analysis of the dwell times between unwinding and re-zipping steps is consistent with multiple ATP binding events occurring for steps >1 bp. From these data, we present a comprehensive kinetic model of UvrD unwinding, re-zipping, and strand release. Structural alignment of the UvrD motor core subdomains with those of SF1 homologs PcrA and Rep, those of the RecB subunit of RecBCD, and those of less closely related 3′–5′ SF2 helicases suggest a conserved stepping mechanism.

## Results

**UvrD unwinds and re-zips dsDNA in ~3 base pair increments.** High-resolution dual-trap optical tweezers[47] were used to probe the stepping behavior of monomeric UvrD. We tethered a single

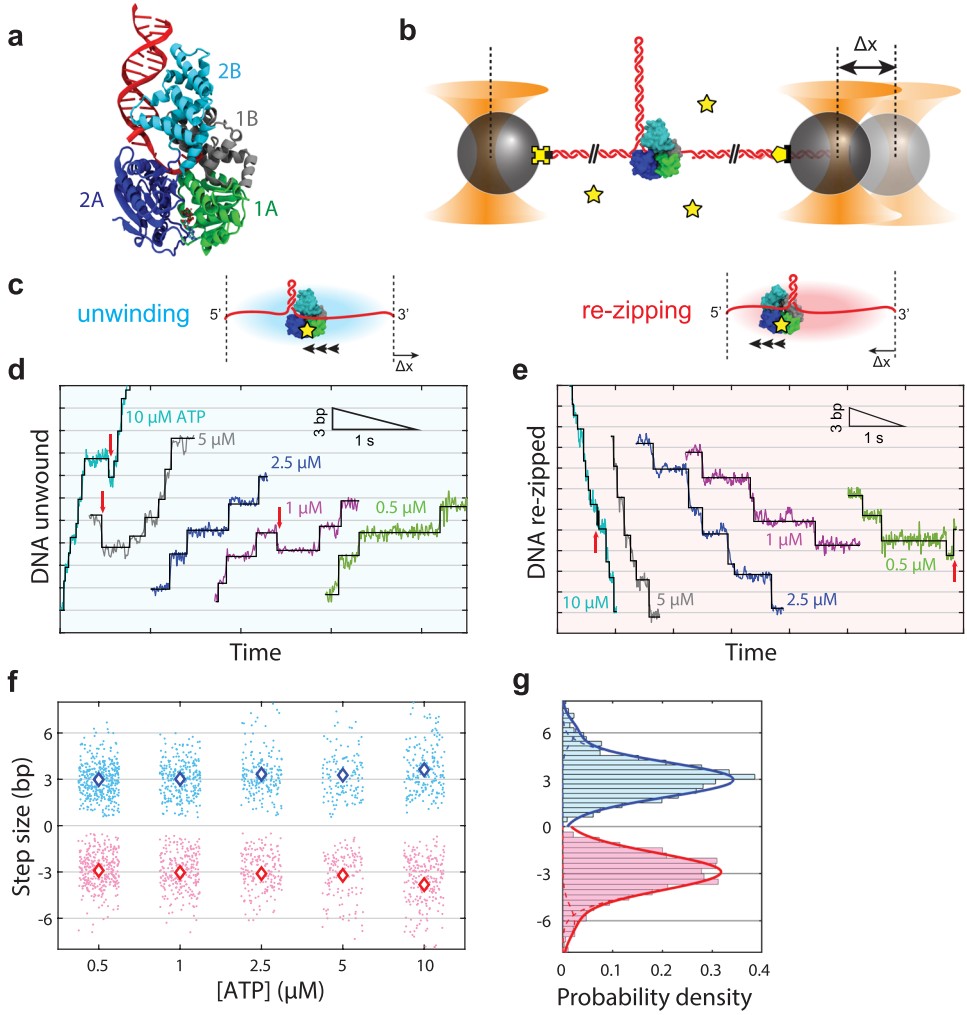

**Fig. 1 UvrD helicase unwinds and re-zips DNA in steps averaging 3 bp. a** Crystal structure of UvrD bound to ss/dsDNA junction (PDB accession number 2IS2) with color-coded subdomains 1A, 2A, 1B, and 2B. **b** Schematic of optical trapping assay. An 89-bp long DNA hairpin stem flanked by two 1.5-kb dsDNA handles is tethered between two optically trapped polystyrene beads by biotin-streptavidin (yellow cross-black square) and digoxigenin-anti-digoxigenin (yellow pentagon-black rectangle) linkages. A 10-dT loading site allows a single UvrD to bind. Addition of ATP (yellow star) after loading UvrD initiates helicase unwinding in a 3′ to 5′ direction, detected as a change in tether extension, $\Delta x$, at constant force. **c** Hairpin unwinding (blue) leads to an increase in extension from the released ssDNA at constant force. Re-zipping (red), mediated by UvrD strand-switching and translocating on the opposite strand of the hairpin stem allowing the DNA to re-zip, leads to a decrease in extension at constant force. **d, e** Representative data traces of UvrD stepping behavior during unwinding (**d**) and re-zipping (**e**) across ATP concentrations (color coded as shown) and forces (11–14 pN). The black lines represent fits to a statistical step detection algorithm. Red arrows indicate backsteps in unwinding and re-zipping traces. **f** Scatter plot of individual unwinding (blue points) and re-zipping (red points) step size measurements at each ATP concentration, and plot of the average step size (blue and red diamonds). Total number of steps displayed: $N = 394, 287, 191, 134, 200$ (unwinding) and 338, 256, 192, 138, 237 (re-zipping) from 25, 16, 12, 12, 12 traces at [ATP] = 0.5, 1, 2.5, 5, 10 μM, respectively. Error bars denoting the s.e.m. are smaller than the diamond symbol size. **g** Histogram of unwinding (blue) and re-zipping (red) step sizes over all ATP concentrations, with fits to double-Gaussian distributions (solid lines; dotted lines represent individual single-Gaussian components). Total number of steps in histogram: $N = 1206$ (unwinding), 1161 (re-zipping) from 77 traces across ATP. Source data are provided as a Source Data file.

DNA hairpin consisting of an 89-bp stem capped with a dT tetraloop and flanked by two 1.5-kb dsDNA handles between two optically trapped polystyrene beads (Fig. 1b; see "Methods"). A $(dT)_{10}$ ssDNA loading site on the 3′ side of the hairpin was used for helicase binding; the 10-nt loading site size guaranteed that a maximum of one UvrD bound to the DNA[48]. All experiments were performed in a laminar flow sample chamber[49] where two adjacent buffer channels merged to form a smooth interface, which allowed for different components to be contained in each channel (Supplementary Fig. 1; see "Methods"). In a typical experiment, a single DNA hairpin tether was formed in situ in the bottom channel containing ATP but no UvrD, moved to the upper channel containing UvrD but no ATP for a 10–30 s incubation period to load the helicase, and finally moved back to

the ATP channel to initiate unwinding. We calculated the number of base pairs unwound over time from the change in extension of the tether as UvrD released the newly formed ssDNA, converting each 2 nucleotides (nt) released into 1 bp unwound (see "Methods"). Data were collected at a constant tension through active force feedback over a range of forces (9–15 pN) below the mechanical unfolding force (~16 pN) of the hairpin (Supplementary Fig. 2), and over a range of ATP concentrations (0.5–10 μM). A typical unwinding trace is shown in Supplementary Fig. 1. In line with previous single-molecule studies[28], monomeric UvrD unwinds short lengths (~20 bp at 10 μM ATP) of dsDNA in short, repetitive bursts. Unwinding is interrupted by periods of gradual DNA re-zipping, which correspond to UvrD translocating on the opposing strand of the hairpin in a 3′ to 5′

direction away from the fork junction, allowing the DNA to reanneal[7,28] (Fig. 1c).

Example traces of UvrD unwinding (Fig. 1d) and re-zipping (Fig. 1e) are displayed across multiple ATP concentrations; in all cases, clear evidence for stepwise motion is observed. UvrD unwinds dsDNA in stepwise increments and displays similar behavior during re-zipping. We note the occurrence of individual backsteps in the opposite direction of both unwinding and re-zipping (<10% of total steps, denoted by arrows in Fig. 1d, e) which we believe are distinct from processive re-zipping (Fig. 1e) and unwinding (Fig. 1d) activity, respectively. As expected for an ATP-dependent translocase operating at conditions where ATP binding is rate-limiting (satisfied here since 0.5–10 μM is well below the reported $K_M$ for UvrD[7]), the dwell times—i.e. the intervals between steps—decrease as ATP concentration increases for both unwinding and re-zipping (Fig. 1d, e).

We used a step detection algorithm[50] to identify stepping transitions in unwinding and re-zipping traces (black lines Fig. 1d, e; see "Methods"), and determined the size of the individual steps from the difference in hairpin positions of the dwells flanking each step. Figure 1f shows a scatter plot of all individual unwinding and re-zipping step sizes, as well as their averages, versus ATP concentration. The average step size is ~3 bp, for both unwinding and re-zipping, largely independent of ATP concentration. We observed slightly larger average step sizes at higher ATP concentrations (e.g. 3.6 and 3.8 bp for unwinding and re-zipping, respectively, at 10 μM ATP), which we attribute to a higher proportion of large steps in the range of 5–6 bp. We believe these larger steps are likely to represent two ~3 bp steps occurring in rapid succession at higher ATP concentrations, which are detected as a single step due to the limited time resolution of our measurement. In support of this interpretation, the fraction of larger steps at each ATP concentration (e.g. 25% at 10 μM) matches well the fraction of short dwell times (<0.015 s or 4 data points at 10 μM) which could be missed during step-fitting. Figure 1g shows the distribution of unwinding and re-zipping step sizes compiled from all steps across ATP concentrations and fits to a double Gaussian (blue and red lines), yielding an average step size of 3.0 ± 1.1 bp for unwinding and −2.9 ± 1.2 bp for re-zipping (mean ± std), recapitulating the results from individual ATP concentrations. (The second Gaussian yields 6.1 ± 0.8 bp and −6.1 ± 1.0 bp, respectively, consistent with double steps.) We corroborated the ~3-bp motor step size with a fitting-independent pairwise distance analysis, using a signed distance to track unwinding vs. re-zipping steps (Supplementary Fig. 3; see "Methods"). Positive and negative pairwise distance distributions for each ATP concentration display a periodicity of 3–4 bp for unwinding and re-zipping independent of ATP concentration.

We also analyzed the effect of force and duplex stability on the step size. Unwinding and re-zipping step sizes were independent of applied force (Supplementary Fig. 4a), and unwinding step sizes showed no correlation with the position on the hairpin stem (Supplementary Fig. 5a). The hairpin sequence has a highly non-uniform G-C content, as manifested in the jagged unfolding transition in the hairpin force-extension curve (Supplementary Fig. 2a) and in the high variance in the probability, $P_{open}(n,F)$, that one or more base pairs downstream of position $n$ in the hairpin open thermally at force $F$ (Supplementary Fig. 2c; see "Methods"). $P_{open}$ exhibits regions of size ~3 bp in the hairpin sequence that have both high (e.g. 5, 21 bp) and low (13, 47 bp) probabilities of melting spontaneously. Our observation that the step size is independent of position indicates that its 3-bp value is unrelated to duplex stability and cannot be the result of spontaneous opening of multiple base pairs. To probe further possible sequence dependence, we measured the step size on an alternative DNA hairpin construct with a more uniform G-C

content. While the global stability of the two sequences is the same (~50% G-C content), the uniform sequence exhibits a low variance in $P_{open}$ and a smooth unfolding transition (Supplementary Fig. 2b, c). Unwinding step sizes at 1 μM ATP for the uniform sequence were ~3 bp, the same as for the original hairpin sequence (Supplementary Fig. 5b).

**Dimeric UvrD also unwinds dsDNA with a ~3 bp average step size.** As the data presented thus far are from monomeric UvrD, which requires force or partner proteins to activate helicase activity[7,28–30], we next measured the stepping behavior of dimeric UvrD. For these experiments, we used a DNA hairpin with a longer ssDNA loading site to allow multiple UvrD to bind (see "Methods"). To confirm the binding of a dimer, we used a confocal microscope integrated into the dual-trap optical tweezers[49] to count the number of dye-labeled UvrD bound to DNA by examination of the total fluorescence intensity and photo-bleaching analysis[28] (Supplementary Fig. 6a; see "Methods"). Supplementary Fig. 6b displays a representative trace of helicase unwinding activity (measured by the optical traps) and, simultaneously, fluorescence intensity (measured by the confocal microscope) showing two photo-bleaching steps, verifying the dimer stoichiometry of the bound UvrD.

We focused our analysis on the unwinding step size of dimeric UvrD, as reliable stoichiometric data from fluorescence could be obtained only during the initial unwinding period of the first burst. Supplementary Fig. 6c shows example traces of dimer unwinding with clear stepwise motion over a range of ATP concentrations. Supplementary Fig. 6d displays a scatter plot of all individual unwinding step sizes and their averages, determined using the same step-fitting algorithm as before, versus ATP concentration. The average unwinding step size for dimeric UvrD is ~3 bp and independent of ATP concentration. Fitting the distribution of step sizes across all ATP to a double Gaussian (Supplementary Fig. 6e) gives an average step size of 3.1 ± 1.0 bp (mean ± std), with the second peak at 5.8 ± 0.7 bp, consistent with double steps. Thus, dimerization does not significantly affect the unwinding step size of UvrD, and all data and analysis presented henceforth are for monomeric UvrD unless stated otherwise.

Overall, our measurements show that both monomeric and dimeric UvrD unwind DNA in 3 ± 1 bp (mean ± std) increments. This unwinding step size is smaller than the kinetic step size estimates of 6 bp and 4 bp determined with magnetic tweezers under conditions in which UvrD was presumed to be monomeric[7] and dimeric[24], respectively. One potential reason for the discrepancy may be the high ATP concentration (>500 μM) in the previous studies, much higher than the range in the current work (0.5–10 μM). However, it should be noted that the study employed an indirect method of determining step size, based on noise analysis of unwinding data, which is subject to large systematic errors. Our results are consistent within experimental uncertainties with previous kinetic estimates of the unwinding step size of dimeric UvrD, which fall in the range ~3–5 bp[6,25,27], and of monomeric UvrD complexed with partner protein MutL, in the range ~3–4 bp[29].

**A variable step size with 0.5-bp periodicity points to non-uniform release of unwound strands.** Inspection of the step size distribution in Fig. 1g reveals a large variation across all ATP concentrations (see "Methods"). We thus inquired if this large variance reflected an inherent variability in UvrD unwinding and re-zipping step size and if 3 bp represents a fundamental stepping unit or an average over a range of possible step sizes. As shown in representative traces across ATP in Fig. 1d, e and in zoomed-in sections at 0.5 μM ATP for monomeric UvrD in Fig. 2a, b, we

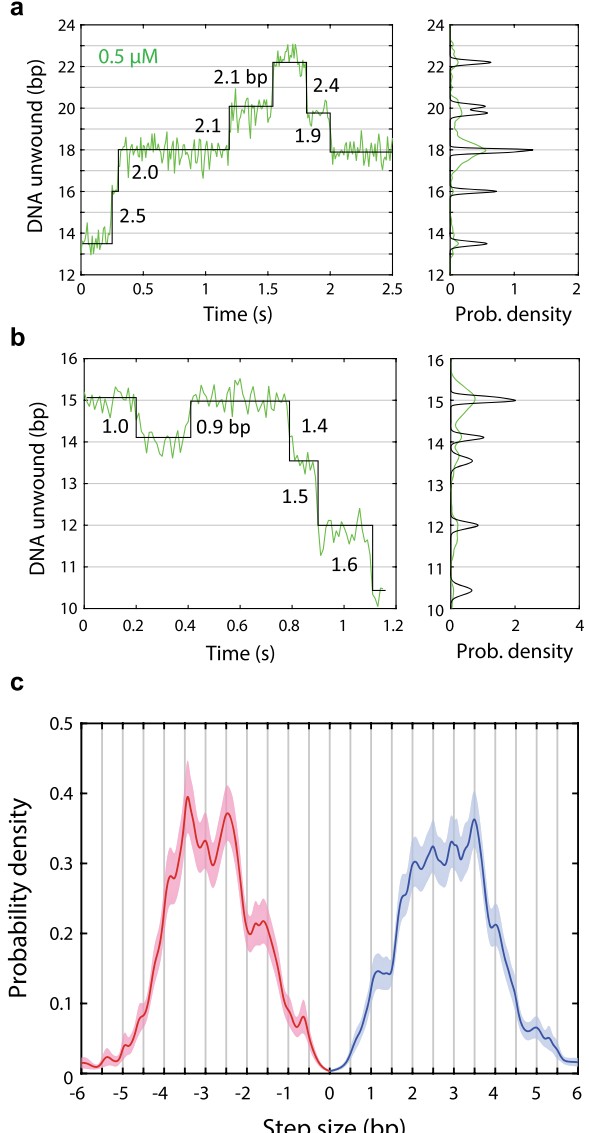

**Fig. 2 The step size of UvrD is highly variable and exhibits 0.5-bp periodicity. a, b** Representative data and step size analysis, highlighting the frequent occurrence of smaller unwinding (**a**) and re-zipping steps (**b**). Example data traces at 14 pN and 0.5 μM ATP (green; left panels) and corresponding dwell position distributions (green; right panels). Fits to steps (black; left panels) and corresponding Gaussian kernels at the most likely dwell position (black; right panels). The kernel widths represent the measurement error for each dwell (see "Methods"). Individual step sizes in base pairs are indicated on the plots (left panels). (**c**) Model of the step size distribution at 0.5 μM ATP. A kernel density estimate of the unwinding (blue) and re-zipping (red) step size distributions based on the dwell position analysis in (**a**) and (**b**) displays 0.5-bp periodicity. Bootstrapping of the kernel density estimate (shaded regions, which denote s.e.m.) confirms the statistical significance of the periodicity in step size distribution. The center of the shaded regions (solid lines) represents the kernel density estimates without bootstrapping and is equivalent to the average of the bootstrapped distributions. All measured step sizes at 0.5 μM ATP and at all forces were used in the construction of the step size distribution. Total number of steps in distribution: $N = 394$ (unwinding), 338 (re-zipping) from all 25 traces at 0.5 μM ATP. Source data are provided as a Source Data file.

commonly observe unwinding and re-zipping step sizes significantly different than the average of 3 bp, with many steps having non-integer values. A gallery of additional unwinding and re-zipping traces at 0.5 μM ATP highlighting the prevalence of non-integer steps and the large variance in step size is shown in Supplementary Fig. 7. Similarly, a gallery of unwinding traces for dimeric UvrD at 1 μM ATP showing non-integer steps and steps smaller than the 3 bp average is presented in Supplementary Fig. 8.

To understand how these smaller steps contribute to the overall step size distribution, we used kernel density estimation (KDE) to obtain a more accurate model of the probability density of step sizes. The KDE was constructed by summing Gaussian kernels for each detected step (right panels, Fig. 2a, b), each centered at the measured step size and with a width equal to the standard error, which we calculated from the measured noise in extension of the dwells flanking each step (see "Methods"). We focused on the lowest ATP concentration (0.5 μM) for UvrD monomers since the dwells flanking the steps are longer under this condition, resulting in lower standard errors for the step sizes. At this concentration, the majority (~80%) of steps have standard errors <0.25 bp, allowing us to differentiate between step sizes <0.5 bp apart. Figure 2c shows kernel density estimates for the unwinding (blue line) and re-zipping (red line) step size distributions. We also carried out a bootstrapping analysis of the KDE distributions (shaded areas in Fig. 2c; see "Methods"), to confirm the statistical significance of features in the distributions.

The unwinding and re-zipping distributions show several statistically significant peaks above and below the 3 bp average. Those below suggest that, although 3 bp is the average step size, it is not the elemental stepping unit. Thus, UvrD exhibits unwinding and re-zipping steps of variable size that combine to give the observed distribution. Notably, many of the peaks in the KDE of both the unwinding and re-zipping step size distributions display a periodicity of ~0.5 bp (Fig. 2c). One potential mechanism for a 0.5-bp step size is that UvrD can unwind a fraction of each duplex base pair per cycle. A non-integer step size has been reported for Hel308 helicase, which displays 0.5-nt sub-steps during ssDNA translocation[37]. However, we believe a more likely scenario is that unwinding of each base pair occurs asynchronously with the release of the two nucleotides generated. The sequestration of unwound ssDNA by UvrD and its subsequent release can explain the 0.5-bp periodicity if an odd number of nucleotides are released, and accounts for the observed variability in step size if varying numbers of nucleotides are sequestered each cycle. Similar delayed strand release mechanisms have been proposed for SF2 helicases HCV NS3 and *E. coli* RecQ[36,41] (see "Discussion").

**Molecular dynamics simulations show that UvrD sequesters unwound DNA strands by looping.** To probe the atomic-level mechanism for the strand sequestration mechanism described above, we carried out enhanced-sampling molecular dynamics (MD) simulations of UvrD bound to a ssDNA–dsDNA junction with extended 3′ and 5′ ssDNA tails (see "Methods"). While the simulations were unable to show dsDNA unwinding, they could reveal the dynamic ssDNA-protein interactions that result in strand sequestration and release. Interestingly, the simulation trajectories demonstrate that both the 3′ and 5′ tails can form loops (Fig. 3a), as a result of non-canonical interactions between ssDNA and the 2A and 1A subdomains. As shown in Fig. 3b, contacts are formed between the ssDNA tails and arginine residues R619 and R213 in subdomains 2A and 1A, respectively. We identified the last nucleotide ID that could form contacts with any

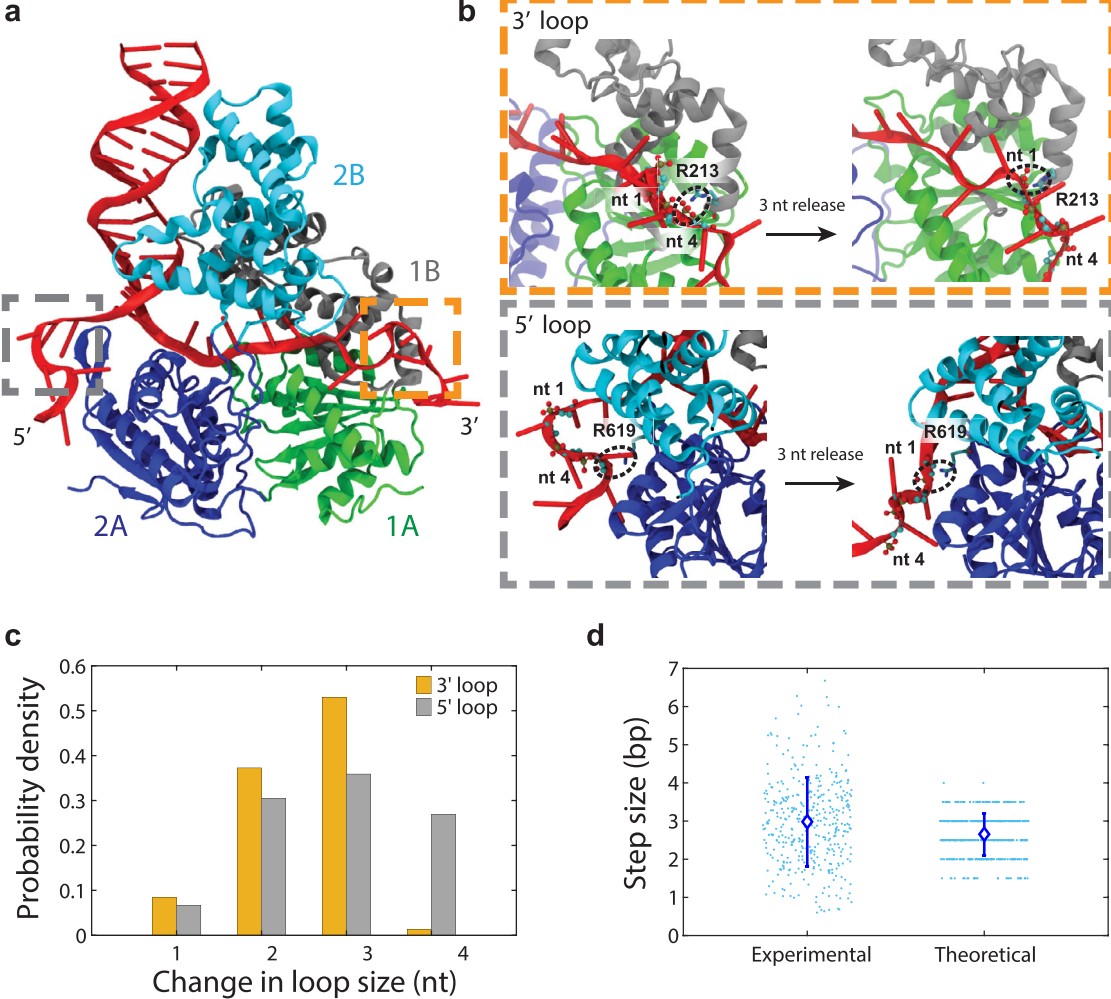

**Fig. 3 UvrD sequesters unwound DNA by looping. a** Snapshot of MD simulation of UvrD-DNA fork complex showing DNA loop formation. Simulations are based on a crystal structure of UvrD (PDB accession number 2IS2) bound to a ss-dsDNA junction with extended 3′ and 5′ tails. Both 3′ (orange dashed box), and 5′ (gray dashed box) loops are shown. **b** Representative conformations showing how the 3′ (orange dashed box) and 5′ (gray dashed box) loops are anchored to the protein surface and released. The primary anchor residues for 3′ and 5′ loops are R213 and R619, respectively. Before the release, nt 4 is in contact with the anchor residue; the 3-nt release is achieved by forming a new contact between nt 1 and the anchor residue. Black dashed circles highlight contact points between the anchor residue and nt 1 or 4. Backbone atoms of DNA phosphate groups and backbone carbon atoms near the contact points are shown in ball-and-stick representation (brown = phosphorous, red = oxygen, cyan = carbon). **c** Distributions of 3′ (orange) and 5′ (gray) loop size changes. **d** Scatter plot comparing the step sizes predicted from simulated loop release to the individual experimental unwinding step size measurements (blue dots) at 0.5 μM ATP, and plot of average step size for both distributions (blue diamonds). Error bars represent standard deviation. Total number of steps displayed: N = 394 (experimental) and 500 (theoretical). Only integer and half-integer values are possible for the simulated step sizes, as they are calculated from distributions in (**c**), assuming loops released from the 3′ and 5′ ends are independent. Source data are provided as a Source Data file.

protein residues for each simulation frame, in order to see how these loops were formed and released during the simulations. The time series of these nucleotide IDs were analyzed to obtain the distributions for the ID change, which represents loop size change. With a 3-nucleotide release being the most likely situation, Fig. 3c shows the distributions of changes in the loop sizes, which range from 1 to 4 nt at both the 3′ and 5′ ends. We next computed a theoretical step size average and standard deviation by compiling the different combinations of 3′ and 5′ loop size changes, assuming that the ssDNA releases at the two ends are independent and follow their own distributions. For example, a 3-bp step size includes all situations where the summation of the 3′ and 5′ release size equals 6 nt. The result (right scatter plot in Fig. 3d) is consistent with the experimental mean and standard deviation in unwinding step size (left scatter plot). The

simulations thus suggest a structural basis for the experimentally observed step sizes.

The most representative conformations depicting the release of 3 nucleotides are shown in Fig. 3b. For the 3′ tail, before loop release the last nucleotide to form contacts with UvrD is nt 4, with R213 being the anchor residue. The 3 nucleotide release is completed after nt 4 separates from R213 and nt 1 forms a stable interaction with R213. For the 5′ tail, before the 3-nt release, nt 4 forms interactions with the anchor residue R619; finally nt 1 becomes engaged with R619. Furthermore, we analyzed the contributions from individual UvrD residues to the non-canonical interactions with ssDNA. The interaction strengths between the ssDNA tail and each protein residue were computed for the frames with the same nucleotide ID. Based on the conformations from simulations, we performed an interaction

energy decomposition calculation and identified four protein residues that had an average generalized Born interaction energy (see "Methods") less than -10 kcal/mol. Residues R213 (−33 kcal/mol) and R96 (−13 kcal/mol) in subdomain 1A formed strong contacts with the 3′ tail, and R619 (−31 kcal/mol) and R331 (−29 kcal/mol) in subdomain 2A were found for the 5′ tail. The main interactions between UvrD and ssDNA are electrostatic and these estimations are comparable to the interaction energies at the primary binding site. These residues are not located in the canonical ssDNA binding site, but to our surprise, they are well conserved especially among such SF1 helicases as PcrA, Rep, and the RecB subunit of RecBCD (see "Discussion").

**Dwell time kinetics reveal that multiple ATPs bind before large steps.** Having identified a mechanism for the UvrD step size, we next analyzed the kinetics of unwinding and re-zipping to elucidate the type and number of kinetic events coupled to motor translocation and ssDNA release. The dwell times between successive steps were determined from the step times identified by the step detection algorithm (Fig. 1d, e; black lines). Since unwinding and re-zipping represent different activities, we reasoned that their kinetics should be separately analyzed. Furthermore, since the dwell time depends on a sequence of kinetic states and their lifetimes[51], and the initial and final kinetic states in a dwell are determined by the preceding and succeeding step, respectively, we classified dwells based on the step direction before and after each dwell. Here, we refer to dwells occurring between two successive unwinding steps as +/+ dwells, and those occurring between two successive re-zipping steps as −/− dwells. Due to their relatively infrequent nature, we did not focus our analysis on +/− and −/+ steps.

We first determined how UvrD kinetics depended on experimental parameters. Dividing the average unwinding and re-zipping step sizes by the respective mean +/+ and −/− dwell times at each ATP concentration, we estimated the unwinding and re-zipping speeds from 0.5 to 10 µM ATP. These estimates were supplemented with measurements of speeds for monomeric UvrD at saturating ATP (100–1000 µM) over the same force range (9–15 pN). At these concentrations, dwell times are too small to allow reliable step detection, and speeds were determined by fitting uninterrupted unwinding and re-zipping periods to a straight line (see "Methods"). Supplementary Fig. 9 displays these speeds versus the full range of ATP concentrations. Fitting the unwinding and re-zipping speeds to Michaelis-Menten kinetics reveal similar kinetic parameters: $V_{max} = 220$ bp/s and $K_M = 39$ µM for unwinding and 210 bp/s and 35 µM for re-zipping. These $K_M$ values are consistent with previously reported values[7], and the maximum re-zipping speed is within the range of prior estimates for ssDNA translocation (190 nt/s) and re-zipping (250 bp/s) from single-molecule and bulk studies[7,20,26]. Interestingly, the maximum unwinding speed is significantly higher than that of dimeric UvrD obtained from bulk studies in the absence of force (70 bps/s)[27], but overlaps with that from single-molecule measurements of monomeric UvrD at high force (200 bp/s)[7]. Similar to the homologous Rep helicase[52], UvrD unwinding speeds appear to increase with force, reaching a maximum equal to the translocation speed above a certain force. Across the measured range (9–15 pN), we observed the average +/+ and −/− dwell durations (and thus the unwinding and re-zipping speeds) to be independent of force, similarly to the step sizes (Supplementary Fig. 4). Dwell durations did not significantly correlate with dwell position on the hairpin (Supplementary Fig. 5). Thus, to facilitate subsequent analysis, we pooled dwell times across all forces and hairpin positions.

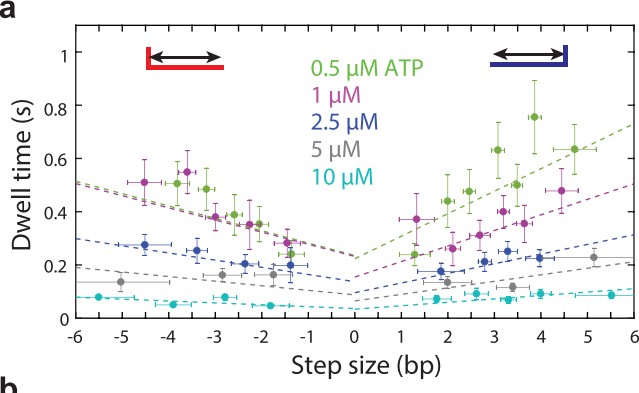

**a**

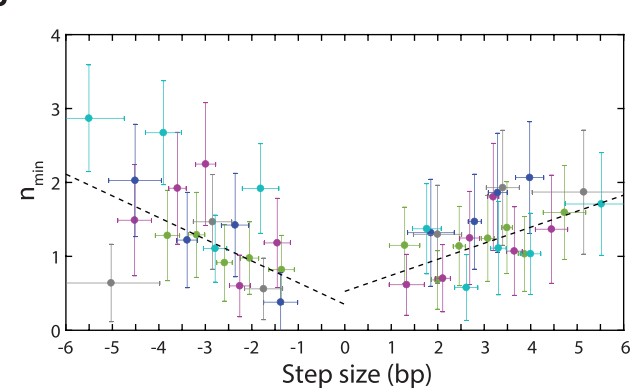

**b**

**Fig. 4 Dwell time statistics reveal that ATP binding and the number of rate-limiting steps depends on step size. a** Average dwell times between successive unwinding (denoted as +/+, shown as positive step sizes) and re-zipping (denoted as −/−, shown as negative step sizes) step pairs vs the size of the step following the dwell, across ATP concentrations (color coded as indicated). Dotted lines represent fits to linear trendlines, obtained by scaling the fits of the normalized dwell times vs step size in Supplementary Fig. 10 by the normalization factors at each ATP. **b** Minimum number of rate-limiting steps, $n_{min}$, for +/+ and −/− step pairs vs the size of the step following the dwell, across ATP concentrations. Dotted lines represent fits to linear trendlines, as described in the text. Vertical error bars denote standard errors and represent the best estimate of the error on mean dwell time and $n_{min}$. Horizontal error bars denote standard deviation and represent the spread in the measured step size (see "Methods"). Total number of dwells contributing to plots in (**a**) and (**b**): N = 214, 156, 113, 84, 153 (+/+) and 162, 128, 105, 83, 193 (−/−) from 25, 16, 12, 12, 12 traces at [ATP] = 0.5, 1, 2.5, 5, 10 µM, respectively. Source data are provided as a Source Data file.

Next, we measured how dwell times depend on step size. Figure 4a shows how the average dwell time depends on the size of the succeeding step, for +/+ and −/− step pairs and across all ATP concentrations where steps were measured. Since the ATP concentration range in our stepping assay (0.5–10 µM) falls well below the measured $K_M \approx 40$ µM for UvrD, we expected ATP binding to be the rate-limiting kinetic event over this data set. Consistent with ATP binding determining the dwell durations, Fig. 4a shows that the mean times decrease monotonically as ATP concentration increases. Moreover, the same plot shows that the average dwell times depend linearly on step size $d$, which suggests that larger steps require either a larger number of ATPs to bind or longer times for ATP to bind. The correlation between dwell time and step size is the same across ATP concentrations. Normalizing the dwell times by their average across step sizes at each ATP concentration, the data at all ATP collapse on the same two trendlines, given by the form $\kappa(d + d_0)$ (with $\kappa = 0.16$ bp$^{-1}$ and

$-0.11\,\mathrm{bp}^{-1}$; $d_0 = 2.6\,\mathrm{bp}$ and $5.1\,\mathrm{bp}$ for unwinding and re-zipping, respectively; see Supplementary Fig. 10).

To distinguish between the two alternatives outlined above (larger steps require either more ATP binding events or longer ATP binding times) we determined and analyzed the variances in $+/+$ and $-/-$ dwell times. The ratio of the mean dwell time squared to the variance, $\mu^2/\sigma^2$, or $n_{min}$, provides a lower limit on the number of rate-limiting kinetic events associated with each dwell ($n_{min}$ equals the inverse of the randomness parameter, used in studying statistical fluctuations in enzymatic reaction dynamics[53]). Plots of $n_{min}$ versus step size for $+/+$ and $-/-$ dwells are shown in Fig. 4b, with fits to linear trendlines of the form $\kappa(d + d_0)$ (with $\kappa = 0.22\,\mathrm{bp}^{-1}$ and $-0.29\,\mathrm{bp}^{-1}$; $d_0 = 2.4\,\mathrm{bp}$ and $1.2\,\mathrm{bp}$ for unwinding and re-zipping, respectively). Across ATP concentrations and dwell type, the plots display similar trends, increasing from ~1 to ~2 with increasing step size, consistent with a greater number of ATPs required to bind before larger steps. Importantly, $n_{min}$ did not grow proportionally with step size (i.e. $n_{min} < 3$ for $d = 3$). Given a 1 nt/ATP chemical step size for UvrD[20], this observation suggests that the multiple ATP binding events preceding a large step cannot all occur at the same rate, but that instead one to two must be slow and rate-limiting.

## Discussion

The measurement and analysis of monomeric UvrD stepping dynamics on a DNA hairpin provide important constraints on its mechanism of unwinding, and molecular dynamics simulations of the helicase-DNA complex point to a structural basis for this mechanism. As depicted in Fig. 5a, we propose that a UvrD monomer moves 1 bp at a time per ATP hydrolyzed during unwinding but sequesters both nascent single strands as loops. As more DNA is unwound, the looped ssDNA accumulates and its delayed release to a shorter loop after a variable number of cycles leads to the measured 3 bp average step size and its high variance. Since the hairpin assay detects ssDNA released, each cycle of single-base pair unwinding does not contribute to a change in extension, while only the release of the sequestered loops results in the discontinuous extension increase responsible for the measured unwinding step size. During re-zipping (Fig. 5b), as UvrD translocates 1 nt per ATP on the opposing strand of the hairpin stem away from the fork junction, we propose that the same sequestration mechanism is in play, except that the loop lengths decrease with every cycle. Here, with each cycle of 1-bp re-zipping, the 2 nt of ssDNA that reanneal are transferred from the sequestered loops back to the hairpin stem, which shortens the loops but does not contribute any change in extension, as the loop tails are bound to the helicase. After a variable number of re-zipping cycles (e.g. 3 cycles for the most probable 3-bp step size) we propose that the initial non-canonical contacts break, allowing new ones to form immediately. These new contacts incorporate several additional nucleotides of stretched ssDNA (e.g. 6 nt for 3 re-zipping cycles) into the motor core to generate a longer loop, producing the discontinuous extension decrease responsible for our measured re-zipping step size. Figure 5c further shows how half-integer unwinding and re-zipping step sizes can result with the respective release or incorporation of an odd number of nucleotides in the 3′ and 5′ loops.

For dimeric UvrD, the nearly identical unwinding step size distribution (Supplementary Fig. 6) and prevalence of non-integer steps (Supplementary Fig. 8) suggests that a similar strand sequestration mechanism may be at play. However, due to the lack of a consensus mechanism of dimer activation, any model of UvrD dimer stepping would be highly speculative at this juncture. While it is known that both helicases in a dimer must be catalytically active[25,27], how each one participates in stepping, strand

sequestration, and loop release remains an open question, meriting further investigation.

Figure 5 represents the model of monomer unwinding most consistent with our data. Strand sequestration is supported by our observation in individual time traces and in step size distributions of half-integer steps (Fig. 2). In addition, our simulations provide a structural basis for delayed strand release through looping (Fig. 3), with the variability in loop length recapitulating the measured step size and its variability. The step size variability (Figs. 1 and 2), coupled with the observation that larger steps are preceded by more than one rate-limiting ATP binding event (Fig. 4), support the notion that multiple of rounds of ATP hydrolysis occur per step and that the number of rounds is variable from step to step. This model also reconciles seemingly disparate step size estimates, as it is consistent with measured ATP coupling ratios of 1 nt/ATP for translocation[20], structure-based models of 1 bp unwound per ATP hydrolysis cycle[17], and kinetic step sizes >1 bp[6,7,24].

In contrast, other models previously proposed to explain non-unitary step sizes (see Introduction) conflict with our experimental results. The dependence of the dwell duration on the succeeding step size strongly suggests that there are multiple hidden kinetic events during dwells, which we attribute to ATP binding and unwinding while the unwound strands remain sequestered. Simultaneous melting models, in which hydrolysis of one ATP leads to unwinding of a variable number of base pairs, are inconsistent with this behavior. (Such models would also result in a coupling ratio >1 bp/ATP). Models in which unwinding occurs in rapid 1-bp steps interrupted by long pauses after a variable number of cycles are inconsistent with this result. Such models would require a pause duration dependent not only on ATP but also on the subsequent number of steps, which is difficult to rationalize. Models invoking ATP-independent base pair melting followed by rapid translocation[45] are similarly at odds with the observed dwell time dependence on ATP and step size. In spring-loaded mechanisms where several base pairs are unwound in bursts following multiple rounds of 1-nt translocation, the multiple translocation steps could represent the hidden kinetic events[40,44] we propose in our model. However, we disfavor such spring-loaded mechanisms (and the other models) as they cannot easily explain the observed half-integer base-pair step sizes.

Our results point to the effect of strand sequestration on UvrD stepping dynamics and suggest a potential mechanism for release of the sequestered strands. The fact that the number of rate-limiting steps, as estimated by $n_{min}$, is less than the number of ATPs we believe must bind for a given step size, 1 bp/ATP, suggests that not all ATPs bind at the same rate. We speculate that ATP binding may become progressively slower as loop length increases during unwinding, and dwell times may be dominated by the slowest 1–2 ATP binding events prior to loop release. Our simulations show that the released loop sizes of ~3 nt are most probable for both the 3′ and 5′ tails, (Fig. 3b), indicating that both shorter and longer loop release is less energetically favorable for unwinding. The structures illustrating the loops represent the most probable loop configurations, indicating that as the loops lengthen during unwinding, they become energetically less favorable than shorter ones. Together, these observations suggest that accumulation of looped ssDNA during unwinding could build up strain, slow UvrD's movement, and eventually trigger loop release and reset of the looping cycle (Fig. 5a). Bending stresses could contribute to strain in these longer loops, with loop release allowing more energetically favorable contacts to form, generating more stable shorter loops.

In contrast, during re-zipping, we speculate that shorter loops are energetically less favorable and that loop shortening generates

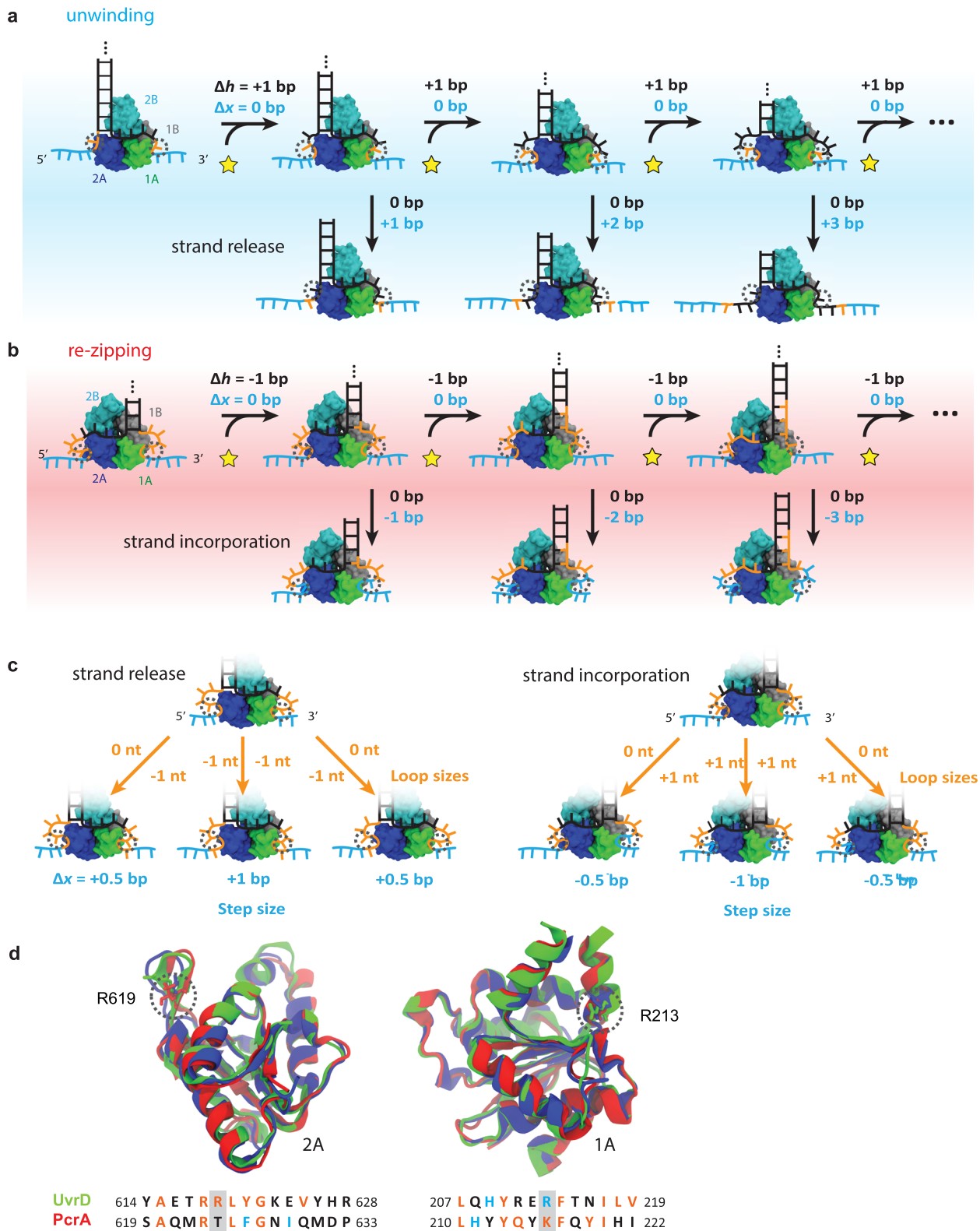

a different type of strain. Here, loop shortening beyond a minimum loop size could stretch loop-protein contacts too far, triggering ssDNA dissociation (Fig. 5b). The next favorable contacts to form would correspond to more energetically stable longer loops (~3 nt longer on average), i.e. a loop incorporation process representing the reverse of the process depicted in Fig. 3b. We note that this model for re-zipping is more speculative, as there

are no crystal structures of the UvrD-DNA complex in the re-zipping state to serve as a basis for MD simulations. Since our past work has demonstrated that the main difference between the unwinding and re-zipping states of UvrD is 2B subdomain orientation[28], we expect the non-canonical binding sites in the 1A and 2A subdomains to play the same roles in the re-zipping state. However, different interactions at the primary DNA-binding site

**Fig. 5 Proposed model for UvrD stepping behavior. a** Model for unwinding. UvrD unwinds 1 bp at a time but accumulates the unwound ssDNA as loops within the motor core that are not released until after later rounds of unwinding, leading to a measured step size >1 bp. The number of rounds of unwinding before strand release can vary, resulting in the variation in step size. DNA is colored to denote different sections of the fork in the initial (leftmost) state and their evolution in each step of the cycle (black = dsDNA hairpin stem and ssDNA bound to the canonical binding site, orange = ssDNA bound as loops, blue = released ssDNA). Loop dissociation in the rightmost state represents the same process depicted in Fig. 3b. **b** Model for re-zipping. UvrD translocates 1 nt at a time on the opposite strand of the hairpin. Each round of re-zipping results in shortening of the loops, with a final release to a larger loop size. In (**a**) and (**b**), $\Delta h$ denotes the number of hairpin nucleotides unwound or re-zipped, while $\Delta x$ denotes the number of nucleotides released. **c** Illustration of loop release and incorporation scenarios responsible for integer and half-integer step size measurements for both unwinding and re-zipping. With uniform strand release during unwinding, one nucleotide is released from both strands (middle), leading to an integer-bp step size. With non-uniform strand release, one strand is released while the other remains sequestered (left and right), leading to a half-integer bp step size. During re-zipping, one additional nucleotide is incorporated into each loop in the uniform process (middle), resulting in an integer-bp step size, while nucleotide addition into one loop in the non-uniform process (left and right) leads to a half-integer-bp step size. **d** Structural alignment of 2A (left) and 1A (right) subdomains of UvrD, PcrA, and Rep, with corresponding sequence alignment (color coding is as follows (see "Methods"): black = not conserved, blue = similar, orange = conserved). Anchor residues R213 and R619 are highlighted throughout by gray dashed circles.

in the re-zipping state[54] likely results in a different geometry for the DNA junction-UvrD interface that could account for differences in looping energetics and accumulated strain between these two states. Although we expect protein-DNA contacts to stabilize energetically favorable loop conformations, fluctuations could lead to spontaneous loop release or incorporation (i.e. unlooping/looping transitions) independent of ATP-catalyzed unwinding. As noted above, we occasionally observe trains of successive unwinding or re-zipping steps that are interrupted by backsteps in the opposite direction (Fig. 1d, e). These events are likely distinct from unwinding and re-zipping activity, and may represent examples of spontaneous unlooping/looping, although other interpretations are also possible.

It is also important to note that our model predicts that multiple different pathways could lead to the same observed step size. For instance, an unwinding step size of 2 bp could result from release of two 2-nt loops from both motor core subdomains, or equivalently from release of 1 nt of one loop and 3 nt of the other (Fig. 5). Thus, several loop release and incorporation pathways likely contribute to the dwell times measured for different step sizes. Since the probabilities and lifetimes of these individual pathways are unknown, it is not currently possible to model unwinding and re-zipping dwell time kinetics more explicitly.

We inquired whether strand sequestration by basic amino acid residues within the motor core subdomains could represent a conserved unwinding and translocation mechanism among SF1 and SF2 helicases. Structural alignment of the motor subdomains in UvrD and its SF1 homologs shows that the key residues involved in loop formation are relatively well preserved. For example, R213 and R619, which in the simulations are key contact residues with the formed loop ends at the 3′ and 5′ tails, respectively, are found to be highly conserved among the 3′–5′ SF1 helicases UvrD, Rep, and PcrA and in the RecB subunit of RecBCD (Fig. 5d and Supplementary Fig. 11). Moreover, structural alignment of the 3′–5′ SF2 helicases NS3 and RecQ (Supplementary Fig. 11), both of which display half-integer unwinding steps and are believed to operate by a strand sequestration mechanism[36,41], reveals positively charged residues in the 1A-2A motor core at similar positions as UvrD/PcrA/Rep/RecB that could serve as anchors for the displaced strand during unwinding. UvrD cannot be aligned well to two helicases shown to exhibit 1-bp step sizes, XPD and Pif1, as they are 5′–3′ helicases.

The physiological role of loop accumulation and delayed release in helicase-catalyzed unwinding remains to be determined. It was previously speculated that strand sequestration could serve a regulatory role to control the speed of unwinding[3], similar to the function of the 2B subdomain in Rep[52,55] and UvrD[25,28–30] in preventing rapid, uncontrolled, and detrimental unwinding activity. Additional studies will be needed to test how

the disruption of looping interactions identified in the MD simulations affect the stepping dynamics and speed of UvrD and related SF1 helicases. All evidence so far suggests that loop-forming interactions are strong. MD simulations estimate interaction energies less than −10 kcal/mol between each anchor residue and the ssDNA tail. The same generalized Born energy approach gives estimates of the interaction strengths of the canonical ssDNA binding site residues between −10 and −40 kcal/mol, comparable to those at the non-canonical binding sites. Moreover, we observe no dependence of step size on applied force over the measurement range (9–15 pN) (Supplementary Fig. 4), similar to NS3 helicase[41], suggesting that the intermolecular forces responsible for loop formation are much greater than the pulling forces applied in our experiments. Nevertheless, it is possible that partner proteins or other DNA-binding proteins could sufficiently disrupt loop-forming interactions and regulate helicase speed and stepping behavior.

## Methods

**Protein expression, purification, and storage**. Wild-type UvrD was expressed and purified from *E. coli*[18,48]. Fluorescently labeled UvrD was synthesized from a UvrDΔCys(A100C) mutant with all naturally occurring cysteine residues replaced with serines and a new, single cysteine introduced at alanine 100, and was labeled by a single AlexaFluore555 dye via maleimide chemistry[18,28]. Proteins were stored at −20 °C in a minimal storage buffer comprised of 50% (v/v) glycerol, 500 mM NaCl, and 20 mM Tris-HCl (pH 8.0) at a concentration of ~800 nM to 1 μM. Concentrations were determined spectrophotometrically using an extinction coefficient $\varepsilon_{280} = 1.06 \times 10^5 \ \mathrm{M^{-1} \ cm^{-1}}$ [48]. Prior to experiments, the UvrD stock solution was first diluted 10-fold in a buffer containing 50% (v/v) glycerol, 200 mM NaCl, and 20 mM Tris-HCl (pH 8.0) and then 100-fold in the experimental solution buffer, giving a final protein concentration of 8–10 nM.

**Experimental solution conditions**. All experiments were conducted at 22 °C in 35 mM Tris-HCl (pH 8.0), 20 mM NaCl, 5 mM MgCl₂, and 2% glycerol in the presence of an oxygen scavenging system to increase the lifetime of the tethers[56] (1.2% glucose, 0.29 mg/mL pyranose oxidase from *Coriolus sp.* (Sigma-Aldrich, St. Louis, MO, USA), and 0.15 mg/mL catalase from *Apergillus niger* (EMD Millipore, Billerica, MA, USA)). Experiments with fluorescently labeled UvrD also included a triplet-state quencher (1 mg/mL Trolox (Sigma-Aldrich, St. Louis, MO, USA) to prevent fluorophore blinking.

**DNA hairpin construct**. The hairpin construct was synthesized[28,35] by ligating a variable 89-bp hairpin stem capped by a (dT)₄ tetraloop to two 1.5-kb double-stranded handles made by PCR amplification of sections of the pBR322 plasmid (New England Biolabs, Ipswitch, MA, USA). The left and right handles were respectively modified with 5′ biotin and digoxigenin to facilitate attachment to streptavidin- and anti-digoxigenin antibody-coated beads. In the final construct, the hairpin insert was flanked on the 3′ side by a poly-dT loading site for helicase binding (Fig. 1b). For experiments probing the stepping behavior of monomeric UvrD, the loading site was 10 nt long to ensure that only a monomer of UvrD bound to the DNA. A longer loading site of 19, 38, or 60 nt was used for experiments examining the stepping dynamics of dimeric UvrD. All oligonucleotides were purchased from Integrated DNA Technologies (Coralville, IA, USA). Sequences for all hairpin inserts and primers are displayed in Supplementary Table 1.

**Optical trapping instrumentation**. Experiments were performed on two custom-built high-resolution dual-trap optical tweezers instruments[47,49]. One of these instruments combines optical trapping with confocal single-molecule fluorescence microscopy[49], and this apparatus was used for all experiments with labeled UvrD. The traps were calibrated according to standard procedures[47], and custom Lab-VIEW code was used for data acquisition. Data at 0.5, 1, 2.5, and 5 μM ATP were collected and analyzed at 100 Hz, while data at 10 μM ATP and higher were collected and analyzed at 267 Hz for a higher temporal resolution to account for the higher unwinding velocity at those concentrations.

**Sample chamber and unwinding assay**. Experiments were performed in a microfluidic laminar flow cell sample chamber[49] for controlled assembly of biomolecular complexes. Chambers consisted of a piece of parafilm patterned with three channels, melted between two glass coverslips (Supplementary Fig. 1). Coverslips were coated with polyethylene glycol (PEG) to prevent protein adsorption to the surface. Chambers were patterned with a central channel comprised of two streams that smoothly merged at a sharp interface, and the laminar flow allowed different buffer components to be contained in each stream without significant mixing. One stream contained ATP (0.5–10 μM) without UvrD, while the other contained UvrD (8–10 nM) without ATP. The top and bottom channels were used to introduce streptavidin- and anti-digoxigenin antibody-coated beads, respectively, into the central channels by means of glass capillaries embedded in the parafilm connecting the top and bottom channels to the central channel. In a typical experiment, a single DNA hairpin was first tethered between two optically trapped beads in the ATP stream and held at constant mechanical tension via an active force feedback mechanism. The tether was then translated into the UvrD stream and allowed to incubate for 10–30 s to load UvrD onto the 3′ loading site before finally moving back to the ATP stream to initiate unwinding (Supplementary Fig. 1). For experiments probing the stepping behavior of monomeric UvrD, a 10-dT loading site was used to ensure that only a single wild-type UvrD bound to the DNA. For measurements of dimeric UvrD unwinding steps, a longer loading site (19–60 dT, typically 38 dT) was used to allow multiple proteins to bind, and fluorescently labeled UvrD was used in conjunction with our fluorescence optical tweezers instrument[49] to determine the number of helicases bound to the DNA. The dimer stoichiometry was verified by counting the number of photobleaching steps and examining the fluorescence intensity to ensure that it was consistent with that of two dyes, as detailed in a previous study[28]. The flow chamber configuration used for these experiments was usually the same as that for the monomer experiments described above, with some exceptions as detailed below. Data for the analysis of monomeric UvrD unwinding speeds at saturating ATP (100–1000 μM) were collected on a modified laminar flow chamber. In this case, the tethered hairpin remained in the top stream containing protein and ATP, allowing UvrD to bind, unwind/re-zip DNA, and dissociate repeatedly. UvrD concentrations were low (~1 nM) to ensure that a single helicase bound the DNA at one time, and the fluorescence signal from labeled UvrD was used to verify monomer stoichiometry. Some of the dimeric UvrD measurements were also carried out on this modified flow chamber.

**Data analysis**

*Analysis of force-extension curves, hairpin sequence stability versus position, and unwinding distance in constant force traces.* Upon forming a DNA tether between two optically trapped beads, a force-extension curve (FEC) was measured to verify that a single, undamaged hairpin was tethered between the beads. The extensible worm-like chain (XWLC) model was used to fit the folded and unfolded regions of the FEC, using the following parameters for ssDNA and dsDNA from a previous study[28]: persistence length, $P_{ds} = 53$ nm and $P_{ss} = 1$ nm, inter-phosphate distance, $h_{ds} = 0.34$ nm/bp and $h_{ss} = 0.59$ nm/nt, and stretch modulus $S_{ds} = 1100$ pN and $S_{ss} = 1000$ pN. The unfolding transition was also fit using the nearest neighbor base-pairing free energies of the hairpin stem sequence[35]. To calculate the number of base pairs unwound over time, the change in extension (in nm) of the DNA construct during unwinding was divided by the extension of the two nucleotides of ssDNA released at the measured force, using the XWLC parameters for ssDNA referenced above. For our analysis of the sequence dependence of UvrD stepping dynamics, we calculated the force dependent probability $P_{open}(n,F)$ that one or more base pairs open thermally downstream of the position $n$ in the hairpin stem at tension $F$, taking into consideration both the base-pairing free energy and the energy of stretching the released ssDNA[35].

*Step size analysis.* Two methods were used to determine the average unwinding and re-zipping step sizes of UvrD: a statistical step detection algorithm based on the method of Kerssemakers et al.[50] and a model-free pairwise distance analysis[53]. In the first method, unwinding and re-zipping segments from bursts of activity were fit using the step detection algorithm, identifying the time points at which stepping transitions were likely to occur. Then, the dwells between stepping transitions were analyzed to determine their most probable positions on the hairpin (in bp). Specifically, a kernel density estimate (KDE) was used to construct the position distribution of each dwell (see, for example, Fig. 2a, b, green curves in right panels), and the most probable dwell position was identified from the peak of the distribution. A Gaussian kernel with standard deviation σ = 0.1 bp was used for KDE of the dwell position distribution. Step sizes were determined from the differences

in position between two consecutive dwells. Step sizes for all traces at a given ATP concentration were combined to determine the step size distributions shown in Fig. 1g. Step sizes <0.5 bp were excluded from the data set. Number of steps, average step size, and step size standard errors at each ATP concentration for monomers are displayed in Supplementary Table 2, while the analogous values for dimer unwinding steps are displayed in Supplementary Table 3.

In the second method, the position differences between every pair of data points were calculated for the same set of traces used in the first method, keeping track of the sign (positive for unwinding steps, negative for re-zipping steps). All the unwinding and re-zipping pairwise distances at a given ATP concentration were then combined to determine the pairwise distance distributions shown in Supplementary Fig. 3.

*Modeling the step size distribution.* To quantify the variability in step size, we generated a KDE of the step size distribution, taking into account the statistical errors on each of the measured step sizes. We treated each dwell position $i$ as a Gaussian kernel, $p_i(x)$,

$$p_i(x) = \frac{1}{\sqrt{2\pi}\sigma_i}e^{-\frac{1}{2}\left(\frac{x-x_i^*}{\sigma_i}\right)^2}, \tag{1}$$

centered at the most probable position $x_i^*$ and with standard deviation $\sigma_i$ equal to the standard error of the dwell position (see, for example, Fig. 2a, b, black curves in right panels).

$$\sigma_i = \sqrt{\left\langle\left(x - x_i^*\right)^2\right\rangle/N_i}, \tag{2}$$

where $N_i$ is the number of uncorrelated data points comprising the dwell. The KDE of the unwinding and re-zipping step size distributions was then constructed from Gaussian kernels centered at each step size, $s = x_{i+1}^* - x_i^*$, and with standard error

$$\sigma_s = \sqrt{\sigma_{i+1}^2 + \sigma_i^2}. \tag{3}$$

In this way, steps with larger measurement errors contributed less to the overall distribution. To generate the final step size KDE, the kernels were summed across all steps and the probability was normalized. To generate Fig. 2c, the kernels were plotted against an abscissa $x$ ranging from 0 to max($s + 3\sigma$) for unwinding steps and ranging from min($s - 3\sigma$) to 0 for re-zipping steps in increments of 0.01 bp.

To estimate the statistical significance of peaks in the step size distribution shown in Fig. 2c, a bootstrap analysis was performed. The set of step sizes was resampled with replacement at random over 10,000 iterations, and a new KDE of the step size distribution was generated for each iteration. The shaded areas in Fig. 2c display the standard errors obtained from bootstrapping and represent confidence intervals in the distribution at each step size value.

*Dwell time and velocity analysis.* Dwell times between successive unwinding and re-zipping steps were calculated from the stepping transition times identified by the step-finding algorithm. Dwell times <20 ms were excluded from the data set at 0.5–5 μM ATP due to the limited temporal resolution of the instrument; 10 μM ATP data were collected on an instrument with a faster acquisition rate, and dwell times <7 ms were excluded. Dwells were classified according to the types of steps occurring before and after each dwell, with dwells flanked by two unwinding steps denoted as +/+ dwells, while dwells flanked by two re-zipping steps are referred to as −/− dwells. +/+ and −/− dwell times were averaged at each ATP concentration. The total number of +/+ and −/− dwells, as well as their average and standard error at each ATP concentration, are displayed in Supplementary Table 2. In the analysis of dwell time vs. step size (Fig. 4a), we plotted the individual dwell times for +/+ and −/− step pairs at one ATP concentration against the size of the subsequent step. We determined the mean dwell time $\langle t \rangle$ and the corresponding mean step size by boxcar averaging over a 25–45-point window. Standard errors were calculated throughout from the standard deviation of the sample divided by the square root of the sample size.

We also analyzed the variance in dwell times. We determined the parameter $n_{min}$, given by the ratio of the squared mean over the variance in dwell times (or the inverse of the randomness parameter)[53], which represents the minimum number of rate-limiting kinetic events comprising each dwell. $n_{min}$ was calculated at each ATP concentration for +/+ and −/− dwells. To determine how $n_{min}$ depended on step size (Fig. 4b), we plotted the square of the individual dwell times against the size of the subsequent step. Boxcar averaging over a 25–45-point window yielded $\langle t^2 \rangle$ vs. step size. Standard errors were calculated from the standard deviation of the sample divided by the square root of the sample size. $n_{min}$ was then determined from the equality

$$n_{min} = \left(\langle t^2 \rangle/\langle t \rangle^2 - 1\right)^{-1} \tag{4}$$

using the mean dwell time for each step size from the above analysis. The standard error in $n_{min}$ was calculated from the standard errors in $\langle t \rangle$ and $\langle t^2 \rangle$.

Bootstrapping analysis of $n_{min}$ was used to identify outliers in the dwell time distributions. Because the variance increases much more rapidly with longer dwell times than the mean squared, we found that outlier long-duration dwells could skew the measured $n_{min}$. In the case of outliers, the distribution of bootstrapped

$n_{min}$ values was multi-modal, with a population of low $n_{min}$ values corresponding to resampled distributions with a high occurrence of outlier dwells. We identified which dwells occurred at a higher frequency in the population of resampled data sets with lower $n_{min}$ values, and removed these outliers from the original distributions, producing more consistent values of $n_{min}$. This routine was used to identify and remove a single outlier dwell from the 2.5 μM ATP unwinding data set and the 1 μM ATP re-zipping data set.

Dwell times were also used to estimate the unwinding and re-zipping velocities of UvrD, by dividing the average unwinding and re-zipping step sizes by the average +/+ and −/− dwell times, respectively. We ignore the contribution of backsteps, represented by +/− and −/+ dwells, to the speeds due to their rarity. Velocities were calculated this way for all data collected at ATP concentrations of 0.5–10 μM, where step size estimates were readily available. For ATP concentrations greater than 10 μM, where identifying steps was not possible due to the short dwell times between steps, unwinding and re-zipping velocities were estimated by fitting manually selected time intervals of uninterrupted unwinding and re-zipping to a straight line. The number of velocity fits, mean velocities, and their standard errors at these higher ATP concentrations can be found in Supplementary Table 4.

**Molecular dynamics simulations.** The UvrD-DNA complex was built based on the crystal structure 2IS2, of which 3′ and 5′ ssDNA tails were manually extended by 8 nucleotides. We first added a stretched 5 nt of ssDNA to the 3′/5′ tails and performed an energy minimization while constraining the protein and dsDNA backbone positions. A similar protocol was then repeated to extend an additional 3 nt ssDNA segment to both tails. The system was solvated in water with 55 mM NaCl (equal to the total experimental monovalent ionic concentration of 35 mM Tris-HCl + 20 mM NaCl) including ~0.15 million atoms in total. All MD simulations were performed using NAMD 2.13[57] with the CHARMM36 force field[58]. Here we used TIP3P water models and non-bonded fix corrections for CHARMM ion parameters[59]. The energy of the system was minimized using the conjugate gradient algorithm. A subsequent 50-ns simulation in the isothermal-isobaric ensemble (NPT) was performed at 1 bar and 310 K while restraining the protein atom positions. Langevin dynamics was applied to maintain the temperature combined with the Nose-Hoover method for pressure control. Bonds involving hydrogen atoms were constrained with the SHAKE algorithm. The Particle Mesh Ewald (PME) method was used for full-system periodic electrostatics and a 12-Å cutoff was applied to non-bonded interactions. To perform the initial equilibration of the system, a 500-ns Gaussian Accelerated Molecular Dynamics (GaMD)[60] was carried out with an RMSD constraint for the protein $C_\alpha$ atoms. For GaMD, we applied boost on both dihedral and total potential energy, of which the upper limit of the standard deviation is set to 6 kcal/mol.

To enhance the sampling of the ssDNA tails, we employed the solute scaling method, which scales the intramolecular potential energy of the solute molecule to lower energy barriers between different conformations and is combined with replica exchange (REST2)[61,62]. The implementation available in NAMD[62] enables the selection of atoms in a hot region of a molecule, whose electrostatic, van der Waals, and dihedral potentials are scaled. REST2 periodically exchanges configurations among a set of $N$ replica systems running at the same time, with the hot region temperature ranging from $T_0$ to $T_N$. Specifically, the charges and Lennard-Jones parameters of the selected atoms are reduced by a factor of $\sqrt{\frac{T_0}{T_i}}$, where $T_0$ is the target temperature and $T_i$ is the effective temperature for replica $i$. An attempt is made to exchange the scaling factors of two neighboring replicas after a certain number of MD steps, and the acceptance of this exchange is subject to detailed balance and the Metropolis criteria. Two UvrD-DNA systems, which used either the 3′ tail ssDNA or the 5′ tail ssDNA as the hot region, were set up for REST2 simulations. Here 20 replicas with a temperature range of 310–450 K were launched in parallel with an exchange frequency of every one picosecond. The initial configurations were obtained from the GaMD simulation mentioned above. Each replica trajectory was simulated for 350 ns in the canonical ensemble, resulting in an accumulated 7 μs simulation time for each system.

As mentioned in Results, to obtain the density distributions for loop size changes, we analyzed the last nucleotide ID that can form contact with the protein. Weighted histogram analysis method (WHAM) was used to reweight the population distributions at different temperature replicas to the target temperature $T_0$ (310 K). The set of frames with the same last nucleotide ID was clustered using the ssDNA tail coordinates and representative conformations were shown in Fig. 3b. As a standard practice, we applied the Generalized Born model[63] to evaluate the interaction strength between the ssDNA nucleotides and protein residues for the simulated conformations. The interaction energy $\Delta E$ between a ssDNA nucleotide and a protein residue is calculated according to $\Delta E = E_{complex} − E_{nucleotide} − E_{protein}$, where $E_x$ is the total electrostatic and Van der Waals energy including the GB solvation energy for system x. A 16-Å cutoff was used for the GB model, the ion concentration was set to 55 mM, and a dielectric constant of 80 for the solvent was used.

The structural images were rendered with VMD[64]. To align the helicase motor domains, we used the MultiSeq program[65] in VMD with default values. The 1A and 2A domains of exemplary SF1 and SF2 helicases were aligned to the corresponding UvrD motor domains separately. The default similarity table in the Boxshade program was used for coloring the residues in Fig. 5d.

**Reporting summary.** Further information on research design is available in the Nature Research Reporting Summary linked to this article.

## Data availability
A reporting summary for this article is available as a Supplementary Information file. Source data are provided with this paper. The source data for Figs. 1–4 and Supplementary Figs. 1–10 are also available in the Illinois Data Bank repository at: https://doi.org/10.13012/B2IDB-5556865_V1. Intermediate structures from the simulations (Fig. 3b), as well as essential input and setup files needed to run the REST2 simulations in NAMD are provided in Supplementary Data 1. The structure deposited with PDB accession code 2IS2 was used to build the initial structure of the UvrD complex. The PDB accession codes for the other helicase structures aligned to that of UvrD are as follows: 1UAA (Rep), 3PJR (PcrA), 5LD2 (RecBCD), 1OYW (RecQ), 1A1V (NS3). Any other raw or processed data supporting the findings of this study are available from the corresponding author upon reasonable request. Source data are provided with this paper.

## Code availability
All experimental data were acquired using custom LabVIEW code. Analysis of experimental data was performed using custom MATLAB code. The LabVIEW code is publicly available at: (https://gitlab.com/chemla-lab-public-code/old-trap-labview-code) and (https://gitlab.com/chemla-lab-public-code/fleezer-labview-code). The MATLAB code is publicly available at: https://gitlab.com/chemla-lab-public-code/2021_natcomm_uvrd_stepping_matlab_codes.

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

## Acknowledgements

The authors thank current and previous members of the Chemla and Lohman labs for helpful discussions, and in particular M.J. Comstock for earlier measurements of stepping dynamics of monomeric and dimeric UvrD. This work was supported by National Institutes of Health grant R01 GM120353 (to Y.R.C.), R01 GM45948 and R35 GM136632 (to T.M.L.), and National Science Foundation Physics Frontiers Center (PFC) program grant PHY-1430124 (Center for the Physics of Living Cells, to Y.R.C. and Z.L.S.). The authors gladly acknowledge supercomputer time provided by the Blue Waters sustained-petascale computing project (OCI-0725070 and ACI-1238993) and by the Extreme Science and Engineering Discovery Environment grant TG-BIO200029.

## Author contributions

S.P.C. and K.D.W. collected and analyzed experimental optical tweezers data. W.M. performed the simulations. H.J. and T.M.L. supplied the protein sample used in experiments. Y.R.C., Z.L.S., T.M.L., and W.M. conceived the research. S.P.C., W.M., and Y.R.C. wrote the original draft of the manuscript. S.P.C., W.M., K.D.W., T.M.L., Z.L.S., and Y.R.C. edited the manuscript.

## Competing interests

The authors declare no competing interests.
