## [Peer Review File · Nature Communications]

Kinetic and structural mechanism for DNA unwinding by a non-hexameric helicaseREVIEWER COMMENTS

Reviewer #1 (Remarks to the Author):

The study focuses on a system of biological importance and provides some new insights into the likely mechanism, which could be of sufficient interest to warrant publication in Nature Comm. My expertise is in molecular simulation and I will confine my comments to that aspect of the study, trusting that other referees will be qualified to comment on the experimental measurements.

The simulations and subsequent analysis have been carefully conducted and to a high standard. They are a minor part of the overall study, providing some suggestive supporting evidence. The authors suggest that the step size predicted from simulated loop release is consistent with experiment, based on the data shown in Fig 3D. The statement is rightly quite measured. Thus, the simulations are a useful addition to the study, but would not be the primary reason to recommend publication in Nature Comm.

The statement and commitment on data sharing could be stronger than "available upon request from the author". Whilst all the simulation data would be too large to share easily, key structures from the simulation and essential setup/input data could be made publicly available, enhancing the reproducibility of the study.

The Generalized Born interaction energies are reported to more significant figures than is justified by the accuracy of the method (the nearest kcal/mol would probably suffice).

Reviewer #2 (Remarks to the Author):

Carney and coworkers employed high-resolution optical tweezers measurements to determine the kinetic step size of UvrD helicase unwinding a short DNA hairpin. The single-molecule measurements reveal a dominant kinetic step-size of 3 bp for unwinding and also re-annealing of the hairpin after UvrD strand switches. The kinetic step-size was independent of ATP concentration, DNA sequence, and the direction of the helicase (unwinding or rewinding). A detailed analysis at limiting ATP revealed that the kinetic step-size is variable and displays half-integer periodicity. Together these findings are consistent with a delayed asynchronous release model in which the helicase unwinds one base per ATP hydrolyzed, but the two single-strands are sequestered and released independently of each other after a certain number of nucleotides are unwound. This mode has been proposed for other helicases. The authors took the significant step of looking for the possible structural basis for the ssDNA sequestration via MD simulations of the helicase with extended ssDNA regions. The results of this analysis suggested non-canonical ssDNA binding by specific residues on the helicase domains, corresponding to the two displaced strands. This is a beautiful and important result that they elegantly expanded through sequence comparisons with other related super family 1 helicases, and structural similarities with super family 2 helicases that share the same mechanistic features. This is an exciting development that provides the first structural model for this intriguing behavior. The authors then probe the kinetics of the individual steps as a function of the kinetic step-size at various ATP concentrations, with the conclusion that the individual kinetic steps associated with ATP binding and turnover are not uniform during the kinetic step, resulting in a single rate limiting step. The authors suggest a model to explain the data.

Overall this is an elegant and important result. The experiments are well done, and the conclusions are well supported by the data. The main result is the proposal of specific residues on the helicase and on other related helicases, that result in the binding and looping of unwound ssDNA that results in the observed kinetic step-size distributions. There is a possibly serious but easily fixed flaw in the proposed model and I have some small points that should be addressed before publication. This work is highly deserving of publication once the points described below are addressed.

1. P6. Dwell time should only decrease with increasing ATP concentration when ATP is limiting.

2. P7. In the discussion of the non-integer steps, I would imagine that the dwell-times between short non-integer steps are possibly much shorter than the dwell-times between full 3 bp kinetic steps. The model would suggest that most of the time both strands are released in quick succession, after a kinetic step of typically 3 bp. Given the fact that the release is typically similar for both strands then I would imagine that instances when one strand is released before the other, the dwell time between the two strand release events is likely shorter than the time between full 3 bp kinetic steps. This fact could complicate the further analysis of the steps since short steps could result from fewer steps before release, or they could represent asynchronous release of the two strands. For example this could impact the analysis and interpretation of figure 4.

3. P9. Can the authors comment on the difference between the unwinding rate of 45 bp/s and the translocation rate of 64 bp/s at 10 μ M ATP. Presumably, the rate is ATP limited at this concentration, and there does not appear to be a strong dependence of the unwinding rate on the stability of the duplex being unwound, which would present one possible reason for the difference between unwinding and translocation rates. Can the authors speculate on the origin of the difference in unwinding and translocation rates? Interestingly, the dwell time as a function of step-size does not appear to vary significantly at 10 μ M ATP but seems to vary to a greater extent at lower ATP concentrations. How do the relative rates of unwinding and translocation compare as a function of ATP concentration? It seems from Figure 4 that the increase in dwell-time with step-size is steeper at 0.5 μ M ATP and is also more pronounced for unwinding than rewinding. Are there possible clues to the difference in the kinetics buried in these measurements?

4. P9. Another way to possibly demonstrate the results in figure 4 is to consider the relative change in dwell time with increase in step-size, or to determine both the slope and intercept of the linear relations between dwell time and step-size. This may be equally as noisy as the ratio of the mean and variance calculations, but could be informative possibly.

5. P 10 and figure 5 B. I am confused by the re-zipping model and how to explain the \sim 3 bp kinetic step for re-zipping. The model for unwinding makes a lot of sense. However, for rewinding the orientation of the UvrD is reversed and I would expect that the extension should decrease smoothly because the extension of the hairpin is dependent on where the "front" of the helicase is with respect to the two strands. My understanding of the discontinuous model is that it imposes an asymmetry between the front of the helicase and the two unwound strands. As the front steps forward in single-nucleotide steps, the two displaced strands remain bound to the helicase for \sim 3 ATPase rounds, at which point they are asynchronously released. Since the measurement of displacement in the tweezers is the separation of the two single strands, this appears as discontinuous increases in extension. When the helicase is resealing the hairpin the asymmetry in the motion seems like it should result in an asymmetry in the resealing reaction since the separation between the two single strands of DNA is set by the location of the front of the helicase, not the location of the two strands that leave the helicase. In this model I would expect to see smooth single nucleotide steps forward associated with each ATP consumed. In the cartoon presented in Figure 5B, I would argue that the apparent extension of the hairpin is completely independent of the number of nucleotides that are looped out behind the helicase, it is dictated instead by where the front of the helicase sits. An alternative model would be one in which the helicase does not act as a reannealing enzyme as shown in the figure, but rather one in which it translocates along ss-DNA and the hairpin re-anneals behind it. In this case the annealed hairpin can only form behind the helicase and the extension change would reflect the same discontinuous motion. There would be one important caveat in this model, which is that unlike the asynchronous release, there could not be asynchronous re-annealing. Rather the reannealing should also be associated with full basepair steps. As soon as the variable amount of ssDNA is released by the enzyme, the free strand can immediately reanneal to it. In this picture the model in figure 5B should be altered to indicate that the hairpin reanneals behind the helicase, and the helicase only interacts with the translocation strand, not the reannealed strand, also the helicase in this case will be translocating along the ssDNA at 90 degrees to the hairpin stem. I think that this model would be in line with the data, and may provide some insight into the increase in reannealing rate, since the helicase is only interacting with one strand rather than two, could the release and reset be faster? As it stands the model in Figure 5 B does not make sense to me. It is possible that I am missing something but if so the description of the model should be clarified – perhaps with a frame by frame depiction of the measured extension below each bp step in the model in 5B.

6. Figure 3. What are the orange dashed circles in the illustration of the MD sims? Also, what do the red and blue dots or chains represent in these figures?

7. Fig 4 – how do the rates of unwinding and translocation vary as a function of ATP

concentration?

This is exciting research that will be of interest among the entire helicase community and will undoubtedly spur additional research. Once the points above are addressed I think that this will be acceptable for publication.

Keir Neuman

Reviewer #3 (Remarks to the Author):

This paper uses dual-beam optical tweezers to study the stepping behavior of individual UvrD monomers on a hairpin substrate. The study revealed variable step sizes for UvrD averaging to 3-bp, and a periodicity around 0.5-bp. The authors proposed a model to account for these observations, where the UvrD monomer sequesters nascent single strands before release. To further support this model, they carried out MD simulations, the results of which suggest that single-stranded DNA may be sequestered by the protein during translocation, in line with the model proposed.

There are several major concerns with this paper. First, the physiological relevance of these results to UvrD helicase is unclear. Among all the helicases people have studied to date, UvrD is perhaps one of the most comprehensively studied, especially in vitro using biochemical methods, spearheaded by the Lohman lab. It has been well-established that UvrD monomer does not have DNA unwinding activity in vitro, first shown by Maluf et al (2003). However the entire paper has exclusively focused on UvrD monomer, an inactive form of the helicase that does not display unwinding activity under typical biochemical assay conditions in vitro. Even though the helicase activity of a UvrD monomer could be activated by mechanical force or by interaction with MutL (Ordabayev et al. 2018), it is potentially dangerous to infer what would happen to the protein or draw broad generalization across different non-hexameric helicases given this major limitation. The predominant 3-bp step size is also at odds with the step size of 4-5bp determined for one of the active forms of UvrD (Ali & Lohman 1997). While the step size were measured using different methods, the differences are not trivial (which will show up as different fits in kinetic measurement) and whether this is related to the differences in a monomer versus dimer, has to be properly reconciled.

The single-molecule data from this paper show significant similarity to what was published previously for full-length NS3 (Cheng et al. 2011) using high-resolution dual-beam optical tweezers: from the type of data presented to the model proposed. However, a key difference between UvrD monomer and NS3 monomer is that while the former does not have measurable helicase activity in vitro, the latter by itself has measurable helicase activity (Serebrov et al. 2009, Jennings et al. 2009), although limited in its processivity.

The substantial development in current manuscript, as compared to Cheng et al. 2011 is the molecular dynamics simulations on protein-DNA interactions that might occur during unwinding. The MD simulations suggest a potential structural basis for sequestration of nascent single strands. These results are interesting however, these theoretical studies have not been subjected to any experimental test. Based on these MD results, residues R213, R96, R619 and R331 might be involved in binding with nascent single strands, and therefore are potential targets for mutations and testing of the MD simulations. How would mutations of these residues change the resulting step sizes measured from single-molecule assay? These are key points to substantiate the current model and advance the mechanistic understanding on this model helicase. However, the current manuscript falls short on this, no experimental validation of these potential contacts was attempted. Positively-charged residues such as R and K are frequent on the surface of DNA-binding proteins, so it is essential to test out experimentally if indeed the residues identified from MD simulations matter in the step size measured.

Third, the key to support a sequestration model is the apparent non-integer step size. To support this model, more traces showing these features need to be presented, in addition to traces in Fig. 2A-B.

Fourth, it is premature to draw conclusions regarding step size dependence on sequence. For the two hairpin substrates investigated, although there are differences in their sequences, both are $\sim 50\%$ GC content. Indeed, the force-extension curves for these hairpins show qualitative similar features of double barrier to mechanical unzipping. Although the authors named these uniform and non-uniform hairpins, these differences, as shown by the data and sequences themselves, are subtle enough to preclude observation of any dependences on sequences.

Lastly, the authors proposed a model where UvrD monomer sequesters both strands, an inevitable outcome from this model is that the duplex DNA would become torsionally constrained throughout translocation and unwinding cycles of the motor protein. Again, the physiological relevance of this is highly unclear: either the protein or the DNA has to rotate in order to release the topological constraint. Experimental test of this is clearly beyond the scope of the current manuscript, but this is a very important point regarding the potential validity of their model, and has to be carefully thought and considered in order for a model of this kind to work.

Minor points:

- (1) The larger step size in the range of 5-6bp is thought to be 3-bp steps in succession, but this needs to be backed up with more rigorous analysis: whether the frequency of these steps are consistent with the missing dwell times expected at these ATP concentrations.
- (2) It is very important to report the number of traces and the total number of samples (N) included in analysis, the authors included these in supplementary Table 2. But perhaps a better way is to include these in figure legends, in particular for Fig. 2c.
- (3) Potential reasons to account for differences in step size measured for monomeric UvrD, which needs to be discussed. This refers to ref. 7. Dessinges et al. 2004.

Reviewer #4 (Remarks to the Author):

The investigators propose that the unwinding occurs in 1 bp steps, but due to non-uniform release of ssDNA, the measured step size appears to be 3 bp. Thus, three catalytic cycles of ATP hydrolysis occur during the time-frame of the release of the ssDNA (figure 5A).

The breakthrough in this manuscript is the identification of two possible anchor sites.

The conclusions from figure 1F and figure 4B seem to be in conflict. Can the investigators address this, perhaps in the section describing ATP dependence.

A difficult aspect of the model to understand is "how does the dissociation of ssDNA occur at the same rate for unwinding as it exhibits for re-zipping"? For unwinding, strand release occurs when the loop is 3 nt, but strand release during re-zipping occurs with the loop of only 1 nt.

Another way to frame this question is "what factors governs the rate of strand release"?

Related, if ATP concentration increases so that the ATPase cycle increases by ~ 5 -fold, does the measured step-size increase? The model would predict an increase because more ATPase cycles can occur during the time-frame for dissociation of ssDNA from the anchor sites. Or, does the rate of ssDNA dissociation from the anchor site somehow increase when ATP concentration increases? If the rate of ssDNA dissociation increases with increasing ATP, what is the physical basis for this conclusion?

Other issues:

Fig 1D. the "backsteps" are not indicated by arrows.

Fig 1E. The axis should be labelled as "DNA reziped".

Response to the reviews

We thank the reviewers for their careful and thorough reading of the manuscript, positive assessment of our work, and their useful criticisms. We are pleased that the reviews recognize our “important result” (Rev. 2) measuring the stepping dynamics of UvrD, the “high standard” (Rev. 1) of the simulations and subsequent analysis, and “the breakthrough [in] the identification of two possible anchor sites” (Rev. 3) in our structural model for UvrD stepping.

We have extensively revised the manuscript to address the key points raised in the reviews. In particular, we have included in our revision additional single-molecule measurements and analysis of:

- 1) UvrD unwinding and rezipping speeds across ATP, which establish the ATP dependence of UvrD dynamics.
- 2) Unwinding by the dimeric form of UvrD, which exhibits the same step size as the monomer.

We believe that these additional measurements provide further insights into the stepping dynamics of UvrD and strengthen our manuscript. These and other textual changes are highlighted in *blue* in the revised manuscript, and detailed in the point-by-point responses to the reviewer comments below.

Reviewer # 1 points:

1. The statement and commitment on data sharing could be stronger than “available upon request from the author”. Whilst all the simulation data would be too large to share easily, key structures from the simulation and essential setup/input data could be made publicly available, enhancing the reproducibility of the study.

We have included with our submission a file that includes all of the source data for each figure. In addition, as requested, we have provided a separate file containing intermediate structures from the simulations (Fig. 3B) along with files needed for the REST2 MD simulations in NAMD.

2. The Generalized Born interaction energies are reported to more significant figures than is justified by the accuracy of the method (the nearest kcal/mol would probably suffice).

We thank the reviewer for this point. We now report these energies to the appropriate number of significant figures and have revised the related text (see p. 9 in the results subsection “Molecular dynamics simulations”).

Reviewer # 2 points:

1. P6. Dwell time should only decrease with increasing ATP concentration when ATP is limiting.

The reviewer brings up an important point. The concentrations over which we measured the step size (0.5-10 μM) are significantly lower than K_M (~40-50 μM), so it would be reasonable to expect dwell times to decrease with increasing ATP concentration for our data. In the revised manuscript, we mention this important point on p. 6 in the results subsection “UvrD unwinds and re-zips dsDNA”.

2. P7. In the discussion of the non-integer steps, I would imagine that the dwell-times between short non-integer steps are possibly much shorter than the dwell-times between full 3 bp kinetic steps. The model would suggest that most of the time both strands are released in quick succession, after a kinetic step of typically 3 bp. Given the fact that the release is typically similar for both strands then I would imagine that instances when one strand is released before the other, the dwell time between the two strand release events is likely shorter than the time between full 3 bp kinetic steps. This fact could complicate the further analysis of the steps since short steps could result from fewer steps before release, or they could represent asynchronous release of the two strands. For example this could impact the analysis and interpretation of figure 4.

This is an important point. Our model predicts that multiple different loop release pathways, both synchronous and asynchronous, could lead to the same unwinding step size. For example, a step size of 2 bp could result from synchronous release of 2 nt loops from both the 3' and 5' loops, asynchronous release of a 4 nt loop from one loop, or synchronous release of 1 nt of one loop and 3 nt of the other. Modelling the detailed kinetics of these processes would require knowing the individual probabilities and rates of all possible pathways for each step size. While it is reasonable to expect dwell times between asynchronous strand release events to be shorter than for synchronous release, as proposed by the reviewer, there is no way of dissecting the different release pathway probabilities from the present study. Thus, it is not currently feasible to devise a complete kinetic model accounting for all of the different pathways. In the revised manuscript, we address this point on p. 13 in the discussion section.

3. P9. Can the authors comment on the difference between the unwinding rate of 45 bp/s and the translocation rate of 64 bp/s at 10 μM ATP. Presumably, the rate is ATP limited at this concentration, and there does not appear to be a strong dependence of the unwinding rate on the stability of the duplex being unwound, which would present one possible reason for the difference between unwinding and translocation rates. Can the authors speculate on the origin of the difference in unwinding and translocation rates? Interestingly, the dwell time as a function of step-size does not appear to vary significantly at 10 μM ATP but seems to vary to a greater extent at lower ATP concentrations. How do the relative rates of unwinding and translocation compare as a function of ATP concentration? It seems from Figure 4 that the increase in dwell-time with step-size is steeper at 0.5 μM ATP and is also more pronounced for unwinding than rewinding. Are there possible clues to the difference in the kinetics buried in these measurements?

To address questions on the ATP dependence of UvrD unwinding and re-zipping, we present a new Supp. Figure 8, which displays the average unwinding and re-zipping speeds of monomeric UvrD across several decades in ATP concentration across forces 9-15 pN. The plots incorporate speed estimates from our stepping data at low ATP (0.5-10 μM) and from additional data collected at high ATP (100-1000 μM), where steps could not be detected due to the high speeds. A fit to Michaelis-Menten kinetics provide values for $V_{max} \approx 200$ bp/s and $K_M \approx 40$ μM , with the latter value in line with a previously reported value

(see Refs. 7). The plots confirm that ATP binding is rate-limiting over the ATP concentration range (0.5-10 μ M) for which stepping data is presented. This new analysis is described in the revised manuscript on p. 10 of the results subsection “Dwell time kinetics”.

Regarding the difference between unwinding and re-zipping rates, we observe a high overlap between the two over the force range assayed (9-15 pN), with moderately faster re-zipping speeds (~10%) on average at low ATP compared to unwinding (Supp. Fig. 8). Previous magnetic tweezers measurements of UvrD by Dessinges *et al.* (Ref. 7) are fairly consistent with this result, with a slightly higher re-zipping speed (~20% across ATP) relative to unwinding. Single-molecule fluorescence-optical trap studies by Lee *et al.* (Ref. 42) report a ssDNA translocation rate of ~200 nt/s, close to that determined in ensemble studies in the absence of force (Ref. 22) and similar to what we observe for re-zipping. However, Lee *et al.* also observe a lower unwinding speed (70 bp/s) for dimeric UvrD at saturating ATP, the same as ensemble studies by Maluf *et al.* (Ref. 43).

These studies together point to an effect of force on unwinding rates due to destabilization of the duplex, the degree to which depends on the DNA fork geometry and the magnitude and direction of the forces applied (different in the above studies). Qualitatively, the results appear consistent with our prior measurements of the UvrD homolog Rep, in which unwinding rates increase monotonically with force and reach values close to the translocation rate above a certain force (see Makurath *et al.* Ref. 52; Fig. 3). Both Rep and UvrD would be designated as mostly “active” helicases, based on previous criteria (see Manosas *et al.* Ref. 37). Over the force range assayed in our study (9-15 pN), we note that dwell times do not depend strongly on DNA sequence or base pair stability (Supp. Figure 5), which is consistent with our observation of similar unwinding and translocation rates over the same force range. We did not extend our current study to forces <9 pN, as the lower resolution prevents the reliable detection of steps. Thus, while the broader question of force dependence of unwinding is interesting, we feel it falls beyond the scope of the current manuscript, which focuses on stepping dynamics.

Finally, we address the last question on the dwell time dependence on step size at different ATP in our response to point #4 below.

4. P9. Another way to possibly demonstrate the results in figure 4 is to consider the relative change in dwell time with increase in step-size, or to determine both the slope and intercept of the linear relations between dwell time and step-size. This may be equally as noisy as the ratio of the mean and variance calculations, but could be informative possibly.

This is an interesting suggestion. In our revised manuscript, we include a new plot (Suppl. Figure 9) showing the relative change in $+/+$ and $-/-$ dwell time (calculated by normalizing each dwell time measurement by the average at that ATP concentration) vs step size. Within the measurement error, the data for all ATP concentrations collapse around the same trendlines, indicating that the dependence of dwell time on step size is the same across ATP concentration. We have also added trendlines in Figure 4A-B, for dwell times vs step size at each individual ATP concentration and for n_{min} vs step size, respectively. Trendlines in Figure 4A were determined from those in Supp. Figure 9, multiplying their slopes and intercepts by the normalization factor at each ATP concentration, which is the average $+/+$ and

-/- dwell at each ATP. The fact that these curves line up well with the experimental data further confirms that the relationship between step size and dwell time is the same across all ATP concentrations.

5. P 10 and figure 5 B. I am confused by the re-zipping model and how to explain the ~3 bp kinetic step for re-zipping. The model for unwinding makes a lot of sense. However, for rewinding the orientation of the UvrD is reversed and I would expect that the extension should decrease smoothly because the extension of the hairpin is dependent on where the “front” of the helicase is with respect to the two strands. My understanding of the discontinuous model is that it imposes an asymmetry between the front of the helicase and the two unwound strands. As the front steps forward in single-nucleotide steps, the two displaced strands remain bound to the helicase for ~ 3 ATPase rounds, at which point they are asynchronously released. Since the measurement of displacement in the tweezers is the separation of the two single strands, this appears as discontinuous increases in extension. When the helicase is resealing the hairpin the asymmetry in the motion seems like it should result in an asymmetry in the resealing reaction since the separation between the two single strands of DNA is set by the location of the front of the helicase, not the location of the two strands that leave the helicase. In this model I would expect to see smooth single nucleotide steps forward associated with each ATP consumed. In the cartoon presented in Figure 5B, I would argue that the apparent extension of the hairpin is completely independent of the number of nucleotides that are looped out behind the helicase, it is dictated instead by where the front of the helicase sits. An alternative model would be one in which the helicase does not act as a reannealing enzyme as shown in the figure, but rather one in which it translocates along ss-DNA and the hairpin re-anneals behind it. In this case the annealed hairpin can only form behind the helicase and the extension change would reflect the same discontinuous motion. There would be one important caveat in this model, which is that unlike the asynchronous release, there could not be asynchronous re-annealing. Rather the reannealing should also be associated with full basepair steps. As soon as the variable amount of ssDNA is released by the enzyme, the free strand can immediately reanneal to it. In this picture the model in figure 5B should be altered to indicate that the hairpin reanneals behind the helicase, and the helicase only interacts with the translocation strand, not the reannealed strand, also the helicase in this case will be translocating along the ssDNA at 90 degrees to the hairpin stem. I think that this model would be in line with the data, and may provide some insight into the increase in reannealing rate, since the helicase is only interacting with one strand rather than two, could the release and reset be faster? As it stands the model in Figure 5 B does not make sense to me. It is possible that I am missing something but if so the description of the model should be clarified – perhaps with a frame by frame depiction of the measured extension below each bp step in the model in 5B.

In Figure 5A depicting our model for unwinding, each 1 bp unwound does not contribute to the change in extension since the two nucleotides generated are transferred from the hairpin stem to the loops interacting with the motor core. Similarly, in Figure 5B depicting re-zipping, the 1-bp re-annealing steps do not contribute any change in extension, as the two nucleotides that re-anneal are transferred from the loops bound to the motor core back to the hairpin stem. For both unwinding and re-zipping, it is always the change in the number of ssDNA nucleotides released from or added to the non-canonical contacts on the motor core that produces the measured change in extension. Thus, for the re-zipping model in Figure 5B, the discontinuous incorporation of several nucleotides of ssDNA into the sequestered loops lengthens the loops, resets each looping cycle (after multiple cycles of ATP hydrolysis), and generates the measured decrease in extension. To clarify this point, we have modified Figure 5B to identify which parts of the

DNA contribute to the measured extension change after each ATP hydrolysis loop reset cycle, and we have also added text in the discussion section (p. 11) elaborating on this aspect of our model.

In our model, we propose that loop dissociation from the motor core is triggered by the build-up of strain, which can occur both as loops lengthen or shorten. During unwinding, nucleotides generated are transferred from the hairpin stem to the loops, lengthening them. We expect larger loops to be less energetically favorable due to bending stresses, eventually leading to their spontaneous release. During re-zipping, the nucleotides that re-anneal are transferred from the loops bound to the motor core back to the hairpin stem, shortening the loops. Here, we expect there to be a minimum loop size, beyond which the loop-protein contacts are stretched too far, triggering dissociation.

The snapshots depicted in Figure 3 represent the most common loop configurations observed in our MD simulations. However, an important point is that our MD simulations are limited to UvrD in the unwinding state, because there is no structural model for the UvrD-DNA complex in the re-zipping state. As a result, the re-zipping model is more speculative. We have assumed that the most probable loop configurations are the same as for unwinding, but this need not be the case. Nevertheless, since our past work (see Comstock *et al.*, Ref. 46 and Ma *et al.*, eLife 2018, Ref. 54) has shown that the main difference between the unwinding and re-zipping state of UvrD is 2B subdomain orientation, the non-canonical binding sites in the 1A and 2A subdomains should play the same role in the re-zipping state. The difference in built up strain between these two states could potentially stem from different interactions at the primary DNA binding site after UvrD switches to the re-zipping state. We have updated the text in the discussion to make the re-zipping model clearer.

6. Figure 3. What are the orange dashed circles in the illustration of the MD sims? Also, what do the red and blue dots or chains represent in these figures?

We thank the reviewer for noticing this omission. The dashed circles highlight the interaction between key loop-forming residues and specific nucleotides in the ssDNA backbone. We have changed the color of the dashed circles to avoid confusion. The red/blue dots and chains represent specific atoms in the nucleotides that interact with these loop-forming residues. We have updated the Figure 3 caption to explain these features.

7. Fig 4 – how do the rates of unwinding and translocation vary as a function of ATP concentration?

As we discussed above, the revised manuscript now includes Supp. Figure 8 which displays unwinding and re-zipping speeds for monomeric UvrD versus ATP concentration for a wide range of concentrations (0.5 - 1,000 μM). Both unwinding and re-zipping follow Michaelis-Menten kinetics, with respective V_{max} and K_M values within the range previously reported in single-molecule studies and bulk measurements. See our response to point #3 for a more detailed discussion of the kinetic parameters. New text describing this new analysis is found on p. 10 in the results subsection “Dwell time kinetics” of the revision.

Reviewer #3 points:

1. First, the physiological relevance of these results to UvrD helicase is unclear. Among all the helicases people have studied to date, UvrD is perhaps one of the most comprehensively studied, especially in vitro using biochemical methods, spearheaded by the Lohman lab. It has been well-established that UvrD monomer does not have DNA unwinding activity in vitro, first shown by Maluf et al (2003). However the entire paper has exclusively focused on UvrD monomer, an inactive form of the helicase that does not display unwinding activity under typical biochemical assay conditions in vitro. Even though the helicase activity of a UvrD monomer could be activated by mechanical force or by interaction with MutL (Ordabayev et al. 2018), it is potentially dangerous to infer what would happen to the protein or draw broad generalization across different non-hexameric helicases given this major limitation. The predominant 3-bp step size is also at odds with the step size of 4-5bp determined for one of the active forms of UvrD (Ali & Lohman 1997). While the step size were measured using different methods, the differences are not trivial (which will show up as different fits in kinetic measurement) and whether this is related to the differences in a monomer versus dimer, has to be properly reconciled.

We feel that there is great value in understanding the stepping dynamics of monomeric UvrD, as this provides a critical baseline from which to investigate the behavior of UvrD “activated” by protein partners. Moreover, in our revised manuscript we present new data exploring the stepping behavior of dimeric UvrD. We utilized a hairpin with a longer loading site to allow the binding of multiple helicases and used fluorescence detection to determine the number bound. Using an instrument that combines high-resolution optical trapping with single-molecule confocal fluorescence microscopy (see Comstock *et al.*, Nat. Meth. 2011), we measured the fluorescence intensity and number of photo-bleaching steps of fluorescently labeled UvrD to verify the dimer stoichiometry. In this manner, we determined the unwinding step size of dimeric UvrD at 1-10 μ M ATP and observed a 3 bp average step size at all concentrations, the same as for monomeric UvrD. We have added text (p. 7, subsection “Dimeric UvrD”), a figure (Supp. Figure 6), and methods section (p. 15-16) detailing these experiments on dimeric stepping behavior.

Although the exact mechanism of activation by dimerization remains unknown and is beyond the scope of the current study, our new result shows that the unwinding stepping behavior is the same for monomers and dimers. Thus, the difference between our measured 3 bp unwinding step size and previous bulk kinetic values of 4-5 bp is not a consequence of different stepping dynamics for monomers and dimers. We suspect that the two measurements are within error of one another. The 4-5 bp step size is determined from an indirect measurement, based on the number of rate limiting events inferred from unwinding kinetics, and has an error of 1 bp or more. We have added text in the results section (p. 7, subsection “Dimeric UvrD”) discussing our measured step size in comparison to other studies.

2. The single-molecule data from this paper show significant similarity to what was published previously for full-length NS3 (Cheng et al. 2011) using high-resolution dual-beam optical tweezers: from the type of data presented to the model proposed. However, a key difference between UvrD monomer and NS3 monomer is that while the former does not have measurable helicase activity in vitro, the latter by itself has measurable helicase activity (Serebrov et al. 2009, Jennings et al. 2009), although limited in its processivity.

The substantial development in current manuscript, as compared to Cheng et al. 2011 is the molecular dynamics simulations on protein-DNA interactions that might occur during unwinding. The MD simulations suggest a potential structural basis for sequestration of nascent single strands. These results are interesting however, these theoretical studies have not been subjected to any experimental test. Based on these MD results, residues R213, R96, R619 and R331 might be involved in binding with nascent single strands, and therefore are potential targets for mutations and testing of the MD simulations. How would mutations of these residues change the resulting step sizes measured from single-molecule assay? These are key points to substantiate the current model and advance the mechanistic understanding on this model helicase. However, the current manuscript falls short on this, no experimental validation of these potential contacts was attempted. Positively-charged residues such as R and K are frequent on the surface of DNA-binding proteins, so it is essential to test out experimentally if indeed the residues identified from MD simulations matter in the step size measured.

Regarding the novelty of our study, we want to highlight that our work on UvrD represents the first direct measurement of the stepping dynamics of a 3'-5' Superfamily 1 helicase (SF1A), independent of the MD simulations presented. As the reviewer correctly points out, UvrD is a very different helicase than NS3 (a 3'-5' SF2 RNA helicase), and yet our results surprisingly indicate similar stepping behavior during unwinding. Two important differences between UvrD and NS3 are the conformational switch in UvrD that leads to strand-switching and re-zipping activity, and the activation of processive unwinding in UvrD by dimerization. Our data show that UvrD surprisingly exhibits the same 3-bp step size during re-zipping and during unwinding as a dimer (the latter new results detailed in our response to Point #1 above). These are significant results on their own, which we expect to be of interest to the scientific community. The MD simulations and structural analysis provide further insights by suggesting a single, unifying mechanism to explain both the unwinding and re-zipping step size and also to explain how disparate helicases may exhibit similar stepping behavior.

We have several points to make regarding the residues proposed to contact the nascent strands. While the reviewer is correct that positively-charged residues are frequent on the surface of UvrD, many are in fact not conserved across 3'-5' SF1 helicases, whereas the four identified in our MD simulations are. Moreover, other positively-charged residues are either located far from the ssDNA tails or show relatively weak interactions with the loops in the MD simulations. Thus, R213, R96, R619 and R331 are the best candidates for forming the DNA loops. As for experimental tests of the MD simulation results, mutagenesis of the loop-forming residues is an ongoing research effort that has thus far proven complex and difficult. We have attempted to mutate all four arginine residues R213, R96, R619 and R331 (to alanine) by introducing each mutation sequentially into UvrD using an *E. coli* expression system. Since the loops make multiple contacts with these residues, we expect it to be necessary to mutate all 4 to see an effect on stepping dynamics. Unfortunately, of the 4 residues, only R96A (the one exhibiting the weakest interaction) has been inserted successfully. DNA sequencing in some cases shows that the mutant picks up an additional mutation at R453, a residue involved in DNA duplex binding. All attempts to insert additional mutations into the R96A mutant have been unsuccessful so far. It is possible that our difficulty mutating these 4 residues is related to their importance to UvrD function, which is essential for *E. coli* viability, although more tests will be necessary. We are continuing to pursue the synthesis of these mutants, but expect this process to take some time. We are reluctant to hold up the current paper further and feel that mutagenesis studies are best left to future endeavors.

3. Third, the key to support a sequestration model is the apparent non-integer step size. To support this model, more traces showing these features need to be presented, in addition to traces in Fig. 2A-B.

As suggested by the reviewer, we have added a gallery of 6 additional traces at 0.5 μM ATP (an equal number for unwinding and for re-zipping) highlighting the variability and non-integer step sizes, Supp. Figure 7.

We would also like to make an important point that perhaps was missed by the reviewer. The kernel density plot in Figure 2C was made using the *entire data set* at 0.5 μM ATP, which includes 394 measured unwinding steps and 338 measured re-zipping steps. Thus, there is no selection bias in the step size distribution, and we are confident in the statistical significance of the non-integer step sizes that emerge from the distribution. We have included the sample size in the caption for Figure 2 (and all other figures) to clarify this point.

4. Fourth, it is premature to draw conclusions regarding step size dependence on sequence. For the two hairpin substrates investigated, although there are differences in their sequences, both are $\sim 50\%$ GC content. Indeed, the force-extension curves for these hairpins show qualitative similar features of double barrier to mechanical unzipping. Although the authors named these uniform and non-uniform hairpins, these differences, as shown by the data and sequences themselves, are subtle enough to preclude observation of any dependences on sequences.

We disagree with the reviewer on this point. While the two sequences possess the same global stability (both have $\sim 50\%$ GC content), they differ significantly in terms of local stability. Previous work by our group (Qi *et al.*, Ref. 29) and others (e.g. Johnson *et al.* Cell 2007, DOI: 10.1016/j.cell.2007.04.038) has shown that local variations in base pairing stability can modulate helicase activity, with GC-rich patches serving as local barriers to unwinding.

We quantify the local base pair stability by calculating the thermodynamic probability $P_{open}(n,F)$ that one or more base pairs downstream of a position n on the hairpin stem opens thermally at a force F . In the revised manuscript, we have added a plot of P_{open} versus hairpin position for the two sequences in question (Supp. Figure 2C). For the non-uniform sequence, P_{open} fluctuates greatly with sequence position, with several significant dips in opening probability (~ 0.2) representing barriers to unwinding. In contrast, the uniform sequence displays a smoother, more constant P_{open} versus position, signifying a lack of significant unwinding barriers. A description of P_{open} has been added to the results section (p. 6, “UvrD unwinds”).

We have also added plots of P_{open} versus position in Supp. Figure 5A-B, where step size and dwell time are plotted versus hairpin position, to highlight the barriers to unwinding in the non-uniform sequence. Supp. Figures 5C-D plot step size and dwell time versus P_{open} for the same sequence, showing that local stability does not significantly affect these quantities, in contrast to other helicases (e.g. see Ref. 29). This observation, together with the similar step sizes and dwell times observed on the uniform and non-uniform sequences (Supp. Figure 5E-F), shows that DNA stability does not significantly impact the stepping behavior of UvrD.

5. Lastly, the authors proposed a model where UvrD monomer sequesters both strands, an inevitable outcome from this model is that the duplex DNA would become torsionally constrained throughout translocation and unwinding cycles of the motor protein. Again, the physiological relevance of this is highly unclear: either the protein or the DNA has to rotate in order to release the topological constraint. Experimental test of this is clearly beyond the scope of the current manuscript, but this is a very important point regarding the potential validity of their model, and has to be carefully thought and considered in order for a model of this kind to work.

While we agree with the reviewer that protein and DNA should rotate relative to each other during unwinding, we disagree that our sequestration model impacts this behavior. The non-canonical protein-DNA contacts responsible for strand sequestration in our model do not impose any new torsional constraints beyond those already present. The topological constraints are imposed by the DNA fork geometry—in our assay, the DNA hairpin and its two strands attached to trapped beads—not by any particular protein-DNA contacts. The reviewer correctly states that the duplex would rotate upon unwinding by UvrD to release any topological constraints, but this is true for this geometry independent of protein. [To illustrate this point, see the work of Inman *et al.* (Nanolett, 2014, DOI: 10.1021/nl503009d), where dual optical traps unwind a DNA hairpin whose stem is attached to a surface. If the stem is torsionally constrained by linking both strands to the surface, then unwinding twists the duplex and increases torsional strain, rendering unwinding more difficult in comparison to that of an unconstrained stem (see Inman *et al.*, Fig. 3).] The only way for the DNA not to be twisted upon unwinding is to cut one strand of the hairpin. Moreover, even if the sequestered loops were to provide additional torsional constraints, we remind the reviewer that the dissociation of these loops from the motor core every ~3 bp would periodically release these constraints so that torsional strain would not accumulate appreciably.

Minor Points:

1. The larger step size in the range of 5-6bp is thought to be 3-bp steps in succession, but this needs to be backed up with more rigorous analysis: whether the frequency of these steps are consistent with the missing dwell times expected at these ATP concentrations.

As suggested by the reviewer, we calculated the fraction of large (>5 bp) positive and negative steps and compared these to the fraction of short ++ and +/- dwell times. Across all ATP concentrations, the fraction of dwells lasting four data points or less for both unwinding and re-zipping match well with the fraction of large steps. Four data points is a reasonable estimate for the duration of short dwells that are bypassed by the step-fitting algorithm, accounting for the larger “double” steps. We have added text discussing this analysis on p. 6 in the results subsection “UvrD unwinds”.

2. It is very important to report the number of traces and the total number of samples (N) included in analysis, the authors included these in supplementary Table 2. But perhaps a better way is to include these in figure legends, in particular for Fig. 2c.

As recommended by the reviewer, we now include the number of traces and sample size N in all of the figure captions, in addition to Supp. Table 2. Furthermore, we include new Supp. Tables 3 and 4 detailing the sample sizes for dimeric UvrD step size analysis and high-ATP speed analysis.

3. Potential reasons to account for differences in step size measured for monomeric UvrD, which needs to be discussed. This refers to ref. 7. Dessinges et al. 2004

The previous estimate of the step size of 6 bp for monomeric UvrD by Dessinges *et al.* (Ref. 7) was obtained at 500 μM ATP, a concentration far higher than the maximum ATP concentration (10 μM) at which stepping behavior was observed in our study. Thus, it is possible that a larger step size was determined due to the differences in experimental conditions. However, it is important to note that the method used by Dessinges *et al.* to estimate the step size was indirect, since the magnetic tweezers instrument in their study did not have the ability to resolve base pair-scale steps. The step size was estimated from a Fourier analysis of the noise in the data, a method which is subject to large systematic errors. We believe that our study provides a much more direct and reliable measurement of step size. Text on the differences between previous measurements of step size and our current work has been added to the results subsection “Dimeric UvrD” on p. 7.

Reviewer # 4 points:

1. The conclusions from figure 1F and figure 4B seem to be in conflict. Can the investigators address this, perhaps in the section describing ATP dependence.

We assume the reviewer is referring to Figure 1F, which plots the step size vs ATP, and Figure 4A, which plots the dwell time vs step size at different ATP concentrations. The data in these two figures are not in conflict with each other. We explain this as follows. At each ATP concentration, we observe a range of dwell times and step sizes (for both unwinding and re-zipping). Figure 4A shows that there is a correlation, on average, between the dwell time preceding a step and the step size at each ATP. The underlying distributions of step sizes at each ATP in Fig. 4A are identical to those in Fig. 1F. If one were to find the average step size at each ATP in Fig. 4A, one would recover those plotted in Fig. 1F, which are independent of ATP. Similarly, if one were to average all dwell times at each ATP in Fig. 4A, one would find that the averages decrease as ATP increases (Supp. Figure 8 uses this result in plotting speed vs ATP).

One source of confusion may be that Fig. 1F shows a scatter plot of *all* individual step size measurements at each ATP concentration whereas Figs. 4A-B do not. Instead, each point in Figs. 4A-B represents a boxcar *average* over a range of step sizes at each ATP concentration, with the averaging window N typically varying between 25 and 45 individual step size measurements. In the revised manuscript, we have added horizontal error bars to the data in Figures 4A-B to emphasize this point. The horizontal error bars are standard deviations, highlighting the spread in step size values for each data point.

2. A difficult aspect of the model to understand is “how does the dissociation of ssDNA occur at the same rate for unwinding as it exhibits for re-zipping”? For unwinding, strand release occurs when the loop is 3 nt, but strand release during re-zipping occurs with the loop of only 1 nt.

Another way to frame this question is “what factors governs the rate of strand release”?

In our model, we propose that loop dissociation from the motor core is triggered by the build-up of strain, which can occur both as loops lengthen or shorten. During unwinding, nucleotides generated are transferred from the hairpin stem to the loops, lengthening them. We expect larger loops to be less energetically favorable due to bending stresses, eventually leading to their spontaneous release. During re-zipping, the nucleotides that re-anneal are transferred from the loops bound to the motor core back to the hairpin stem, shortening the loops. Here, we expect there to be a minimum loop size, beyond which the loop-protein contacts are stretched too far, triggering dissociation. The snapshots depicted in Figure 3 represent the most common loop configurations (which have lengths of 4 nt and 1 nt) observed in our MD simulations. The simulations further show that the most likely *changes* in loop size are a net decrease of 3 nt (unwinding) or increase of 3 nt (re-zipping). The schematics in Figure 5 synthesize this structural information with this model. During unwinding, the loop length increases from 1 nt, and as it exceeds 4 nt the loop releases back to nearest stable loop length of 1 nt. During re-zipping, the loop length decreases from 4 nt, and below 1 nt the loop dissociates, snapping back to a 4 nt length. (Figure 5 also shows alternate, less probable pathways to illustrate the variability in step size.) We have revised the schematic in Figure 5 to illustrate more clearly this model, delineating which portions of the DNA are responsible for extension changes measured by the optical traps.

One important point to make is that our MD simulations are limited to UvrD in the unwinding state, because there is no crystal structure for the UvrD-DNA complex in the re-zipping state. As a result, the re-zipping process is more difficult to visualize than unwinding, and the corresponding model is more speculative. Here, based on our past work (see Comstock *et al.*, Ref. 46 and Ma *et al.*, eLife 2018, Ref. 54) which showed that the 2B subdomain orientation is the main difference between the unwinding and re-zipping state for UvrD, we have assumed that the most probable loop configurations are the same as for unwinding. However, this need not be the case, as the protein-DNA interactions at the primary DNA binding site will be different in the re-zipping state, which could possibly account for differences in built up strain between the two states. We have updated the text in the discussion to make this point and the re-zipping model clearer.

3. Related, if ATP concentration increases so that the ATPase cycle increases by ~ 5-fold, does the measured step-size increase? The model would predict an increase because more ATPase cycles can occur during the time-frame for dissociation of ssDNA from the anchor sites. Or, does the rate of ssDNA dissociation from the anchor site somehow increase when ATP concentration increases? If the rate of ssDNA dissociation increases with increasing ATP, what is the physical basis for this conclusion?

Our data show that the step size distribution does not change with the duration of the ATPase cycle. We refer the reviewer to Figure 1F, where we show that the average step size for both unwinding and re-zipping remains ~3 bp over a 20-fold range in ATP concentrations, from 0.5 to 10 μM . This range of ATP concentrations is significantly lower than the reported value of K_M (~40-50 μM), indicating that ATP

binding is rate-limiting. Thus, decreasing ATP concentration increases the average dwell time duration (as shown in Fig. 4A) and decreases the speed (as shown in new Supp. Fig. 8) over the range of concentrations we explored.

The reviewer suggests that our model would predict that decreasing the duration of the ATPase cycle would increase the step size by allowing more cycles to occur before ssDNA dissociation. However, we suspect that other factors related to the stability of the sequestered loops are likely to drive release. As stated in our response to point #2, we propose that loop accumulation builds up strain. After a certain number of unwinding cycles, loops become too large, building up strain, and triggering loop release. Thus the step size would depend on loop size, not on the duration of the cycle.

Minor Points:

Fig 1D. the “backsteps” are not indicated by arrows.

Fig 1E. The axis should be labelled as “DNA rezipped”.

These issues have been fixed in the revised manuscript.

REVIEWER COMMENTS

Reviewer #1 (Remarks to the Author):

I raised only minor comments in my review, which have been addressed satisfactorily. My review was confined to the simulations, which are a relatively minor component of the study. The other referees will, presumably, comment on the responses to their reviews.

Reviewer #2 (Remarks to the Author):

The authors have largely addressed my comments in the revised draft. However, I remain skeptical of the model they are presenting for hairpin rewinding in Figure 5 of the revised manuscript. I maintain that there is a fundamental asymmetry in the unwinding versus rewinding process in relation to the two single-strands of DNA and how they bind to the putative ssDNA binding sites on the helicase. When unwinding, the direction of translocation will increase the length of the looped regions until one or both loops become unstable and releases. During rewinding the motion of the helicase would be expected to decrease the size of the loop. Once the loop releases it seems as if the model postulates that the ssDNA will spontaneously loop out as it re-binds to the two putative ssDNA binding sites. Another way to phrase the question is by asking how the rewinding process transitions from the right hand panel of figure 5B back to the left hand panel of 5B? Rather than forming a loop it seems that the rewinding process should tend to decrease the loop size, which is depicted in the panels moving from left to right in figure 5B. My question is how is the loop formed in the first place. After the initial loop is released, what is the process through which the loop is formed during rewinding? It appears as if the loop formation is happening spontaneously after release of the ssDNA from the putative binding sites. If this is the proposed model then the authors should clarify this subtle point - that the discontinuous steps in re-zipping occur due to binding of ssDNA loops before translocation takes place. This was not obvious to me. Though I remain skeptical that this is how the rewinding steps occur, I would be okay if the authors simply clarified the rewinding model to make it more clear. One additional point is that if I am interpreting the model correctly, the steps in rewinding require a thermal fluctuation of the ssDNA to loop out ssDNA at the two ssDNA binding sites on the helicase to produce the step. In this case there is the formal possibility that increased force may decrease the probability of forming a loop or decrease the size of the loops formed. It is likely that under the force range considered the probability of forming a 4bp ssDNA loop is not significantly altered, but it would be a good sanity check on the model to explicitly calculate the looping probabilities as a function of force to show that they do not depend appreciably on force over the experimental range, in accordance with the independence of the step-size. Additionally, it seems that this model should also result in noisier steps with occasional back steps since a large loop (3-4 nt) may spontaneously release and be replaced by a smaller loop (1-2 nt). Finally, if the putative ssDNA binding regions can spontaneously bind looped out DNA in the rewinding mode, then it seems that nothing is preventing them from doing the same in the unwinding mode, though this would result in apparent backsteps of the enzyme. For these reasons I am skeptical of the proposed rewinding model, but if the authors clarify the model and address some of the potential issues with it in the discussion, then I would be satisfied and would recommend publication.

Reviewer #3 (Remarks to the Author):

This paper is interesting in that it shows significant similarity at mechanistic level between UvrD monomer and NS3 helicase that has been extensively studied previously in the literature of helicase mechanisms. Given the structural similarity between these two proteins (even though sequence level identity is low), the similarity at mechanistic level is not totally unanticipated. The single-molecule data are of high quality and thus I favor the publication of this manuscript. However, there are two very important points in my original review that should be further

explored: one on the functional form of UvrD and one on the sequence dependence.

To help the field moving forward, here are my suggestions based on the data in this revised form:

(1) It should be made clear to readers that UvrD monomer does not have detectable helicase activity in vitro under bulk conditions.

(2) The data on UvrD dimer shows 3-bp step size, which is interesting. However, the analysis is incomplete. If the data are sufficient in quantity and high in quality, I suggest the authors to conduct the same analysis as they did for UvrD monomer to examine if there are occurrences of noninteger steps. If so, does it support the sequestering mechanism? This analysis is highly desired because it could place the sequestering mechanism in the context of a functional species that is activated.

(3) The sequence dependence is a contentious topic. The face value of the data in Supp. Fig. 5 is no apparent dependence on local sequence barriers, however, the interpretation of the data needs more caution. If the step size of the helicase is simply 1 bp and no other complex mechanism(s) is involved to yield an apparent bigger step size, I agree that the unwinding activity shows sequence independence. However, since the step size shows a peak at 3 bp, and in order to detect the sequence dependence of the unwinding activity, the local sequence barrier should be 3-bp or bigger. It is not fair to draw conclusions based on the presence of these local barriers that may not be sufficient at all to see the dependence of unwinding dynamics on sequence. Instead, barriers of different length should be designed. An alternative way to examine this is to study unwinding of AT base pairs and compare that with unwinding of GC base pairs, which is a practice by many biochemists in this field. The comparison at single-molecule level will be certainly more revealing: how the dwell may change with sequence and how the apparent step size may also change with the sequence et al.

(4) Lastly, regarding the discrepancy on step size measured for UvrD dimer, the question is that whether 3-bp on average is different from 4-5 bp measured using pre-steady state method. Instead of stating 4-5 bp result 'subject to large uncertainties'. It would be better to refit the published ensemble data using a kinetic mechanism that incorporates what's known, for example, to constrain the step size to 3 bases, but add additional steps that are known to occur. One obligate event in quenched flow measurements is strand dissociation, because only complete unwinding is detected and in reality there is a final step of strand dissociation. This was not included in the original analysis but now can be incorporated and determine whether the original ensemble data may also be consistent with a 3-bp mechanism. To put it in a simple way, given so much more is known about UvrD dimer now than back in 1997, can one construct an alternative kinetic mechanism that can adequately describe the data, incorporating all the knowns with a step size of 3 bases?

- Wei Cheng

Reviewer #4 (Remarks to the Author):

The authors provide thorough and complete responses to all of my questions and concerns. Furthermore, extensive explanations are provided for the other reviewer's comments.

Response to the second round of reviews

We thank the reviewers for their additional helpful comments on the manuscript and their positive assessment of the “high quality” (Rev. 3) of the single-molecule data. We have incorporated new revisions to address these additional points, the three most important of which are as follows:

- 1) Clarifying the re-zipping model for UvrD, emphasizing key differences between the unwinding and re-zipping states and further explaining the loop incorporation process in our re-zipping model.
- 2) Further analysis of the step size data for dimeric UvrD at low ATP, highlighting the prevalence of steps smaller than the 3 bp average and non-integer steps for the dimer.
- 3) Clarifying our discussion on the dependence of step size and dwell times on hairpin position and DNA sequence.

These major changes are detailed in our point-by-point responses to the reviewer comments below. Our revision also contains corrections to minor errors (e.g. Supplementary Table 1, which previously provided an incorrect DNA sequence for one primer). All textual changes are highlighted in *red* in the revised manuscript to distinguish them from the changes made for the first revision in *blue*.

Reviewer # 2 points:

1. The authors have largely addressed my comments in the revised draft. However, I remain skeptical of the model they are presenting for hairpin rewinding in Figure 5 of the revised manuscript. I maintain that there is a fundamental asymmetry in the unwinding versus rewinding process in relation to the two single-strands of DNA and how they bind to the putative ssDNA binding sites on the helicase. When unwinding, the direction of translocation will increase the length of the looped regions until one or both loops become unstable and releases. During rewinding the motion of the helicase would be expected to decrease the size of the loop. Once the loop releases it seems as if the model postulates that the ssDNA will spontaneously loop out as it re-binds to the two putative ssDNA binding sites. Another way to phrase the question is by asking how the rewinding process transitions from the right hand panel of figure 5B back to the left hand panel of 5B? Rather than forming a loop it seems that the rewinding process should tend to decrease the loop size, which is depicted in the panels moving from left to right in figure 5B. My question is how is the loop formed in the first place. After the initial loop is released, what is the process through which the loop is formed during rewinding? It appears as if the loop formation is happening spontaneously after release of the ssDNA from the putative binding sites. If this is the proposed model then the authors should clarify this subtle point - that the discontinuous steps in re-zipping occur due to binding of ssDNA loops before translocation takes place. This was not obvious to me. Though I remain skeptical that this is how the rewinding steps occur, I would be okay if the authors simply clarified the rewinding model to make it more clear.

The reviewer is correct in stating that helicase translocation will decrease the size of the loops in our model of the re-zipping process. We propose that loop shortening generates strain, and that, once a minimum loop size is reached, the protein-DNA loop contacts are stretched too far, triggering dissociation. Once this smaller loop is released, a new, larger loop will form through re-binding of the stretched ssDNA to the non-canonical protein contacts, resulting in discontinuous re-zipping steps.

The reviewer asks why a longer loop is formed. We wish to point out that there are key asymmetries between the unwinding and re-zipping states. First, loops shorten during re-zipping, which triggers dissociation of short loops and thus would favor formation of longer loops. Second, differences in the DNA junction-UvrD geometry between the unwinding and re-zipping states are also likely to lead to an asymmetry in the UvrD-DNA loop binding interactions. As shown in Fig. 5A & B, during unwinding the 3' tail of the ss/dsDNA junction is bound to the primary binding site, while for re-zipping it is the 5' tail, leading to different interactions with the fork. We propose that these differences could cause longer loops to be more energetically favorable in the re-zipping state in contrast to shorter ones being favored in the unwinding state. In the unwinding state, our simulations show that the most probable last contact point for the loop is nt 1 (see Fig. 3B) corresponding to a short loop. As we mentioned in our previous response, the re-zipping model is more speculative because there are no crystal structures for the UvrD:DNA complex in the re-zipping state. Our previous work (Ma *et al*, eLife, 2018) suggests how the UvrD-DNA fork interface may differ between the two states, but simulations of the re-zipping state and a quantification of looping interactions in this state has to-date not been possible.

We have revised and added text in the discussion (p. 13 and 14, paragraphs 3 and 1-2, respectively) to make these key points on the re-zipping model clearer. We highlight the idea that different UvrD-DNA fork interactions could account for the difference in looping energetics between unwinding and re-zipping, and emphasize explicitly our proposal that longer loops may be less energetically favorable during unwinding, while shorter loops may be energetically disfavored for re-zipping. We also make it more evident that, during re-zipping, dissociation of an energetically costly smaller loop is immediately followed by formation of new protein-DNA contacts to form a more stable larger loop, representing the reverse of the loop release process that occurs for unwinding. In addition we have revised Fig. 5A & B so that changes in the loops, released DNA, and unwound hairpin at each step in the kinetic cycle are clearer.

2. One additional point is that if I am interpreting the model correctly, the steps in rewinding require a thermal fluctuation of the ssDNA to loop out ssDNA at the two ssDNA binding sites on the helicase to produce the step. In this case there is the formal possibility that increased force may decrease the probability of forming a loop or decrease the size of the loops formed. It is likely that under the force range considered the probability of forming a 4bp ssDNA loop is not significantly altered, but it would be a good sanity check on the model to explicitly calculate the looping probabilities as a function of force to show that they do not depend appreciably on force over the experimental range, in accordance with the independence of the step-size.

As stated in our response to question #1, we expect loop formation to be energetically downhill for re-zipping. Protein-DNA contacts will stabilize the loops and compensate for any free energy increase due to looping itself. The protein-DNA interaction energies calculated from our MD simulations for the loop contacts are large, comparable to those for the canonical binding site, indicating strong interactions. Thus, if we are interpreting the reviewer's comment correctly, we disagree that thermal fluctuations are required to produce a step, insofar as loop formation is energetically favorable for re-zipping due to these protein-DNA contacts.

Perhaps the reviewer is referring to a different scenario, in which spontaneous loop formation due to thermal fluctuations is first necessary *before* the loop is stabilized by protein contacts. We may then think of the spontaneously formed loop as the transition state to forming thermodynamically favorable protein-DNA contacts.

To see if such a mechanism is plausible, we carried out a calculation on spontaneous looping against force. Using the simple model outlined by Phillips et al. in their textbook “Physical Biology of the Cell” (Ch. 10), we estimated the free energy of looping, to which we added the contribution of force. We note that due to the high flexibility of ssDNA, the elastic energy cost of looping is ~100-fold lower for ssDNA compared to dsDNA of the same length, making spontaneous loop formation much more likely. For an average loop size of 3 nt, this model yields a free energy of looping of ~10 $k_B T$ (~5.9 kcal/mol). Looping these 3 nt against an applied force in the range of 9-15 pN, incorporated into this model by a Fx work term and stretching free energy, contributes an additional ~1.5-3 $k_B T$ for a total energy of ~11.5-13 $k_B T$ (~8.2 kcal/mol). The values estimated by this simple model are quite reasonable for such a barrier height, comparable to those found in ssDNA-binding proteins (see for example Suksombat et al. eLife 2015). Thus, thermal activation over a fully looped transition state is plausible, although we favor mechanisms in which the barrier to loop formation would be low.

The effect of force contributes at most 25% of the total loop free energy, and therefore should not have a dramatic effect on activation over the barrier. Once stabilizing protein-loop contacts are made, we believe the intermolecular forces involved are thus much stronger than the pulling forces, consistent with the independence of unwinding and re-zipping step sizes on force over the applied force range (9-15 pN).

These points on the force independence of the step size and the strength of the loop forming interactions are discussed in the last paragraph of the discussion (p. 15) in both the original version of the manuscript and its subsequent revisions.

3. Additionally, it seems that this model should also result in noisier steps with occasional back steps since a large loop (3-4 nt) may spontaneously release and be replaced by a smaller loop (1-2 nt). Finally, if the putative ssDNA binding regions can spontaneously bind looped out DNA in the rewinding mode, then it seems that nothing is preventing them from doing the same in the unwinding mode, though this would result in apparent backsteps of the enzyme.

Our answers to questions #1 and #2 above address the question of spontaneous loop formation. At the lower ATP concentrations assayed (0.5-1 μM), we sometimes observe repetitive patterns of forward and backward steps (see representative data trace in Figure R1 below). These are unlikely to correspond to strand-switching events and may instead represent examples of looping/unlooping. It is possible that such events are detected at low ATP because translocation is slowed considerably or paused, allowing the time for multiple such looping/unlooping transitions to occur while UvrD is at one position on the fork.

Moreover, we occasionally see negative and positive “backsteps” that interrupt trains of successive unwinding and re-zipping steps, respectively (see revised Figure 1D & E, red arrows). We speculate that these backsteps can also be attributed to ATP-independent looping/unlooping. Unfortunately, both the repetitive hopping and interruptive backsteps are too rare to provide a reliable analysis of their statistics, which could be informative for understanding looping dynamics.

In our second revision, we have included text on spontaneous loop formation/release and supporting evidence in our data in the discussion section (p. 14, paragraph 2).

Figure R1: Repetitive hopping at low ATP concentrations. Sample time trace at 0.5 ATP illustrating UvrD repetitively stepping between two positions in the hairpin stem likely as a result of looping and unlooping. Histograms represent the distribution of positions for the example trace.

Reviewer # 3 points:

1. It should be made clear to readers that UvrD monomer does not have detectable helicase activity *in vitro* under bulk conditions.

We already stated in both the introduction (p. 4, last paragraph) and results section (p. 7, paragraph 2) that UvrD monomers require the application of force to the DNA or the presence of partner proteins to activate helicase activity. Nevertheless, we have added text in our second revision earlier on in the introduction (p. 3, paragraph 4) stating that dimers are required for helicase activity *in vitro*, with monomer activated by force or accessory proteins.

2. The data on UvrD dimer shows 3-bp step size, which is interesting. However, the analysis is incomplete. If the data are sufficient in quantity and high in quality, I suggest the authors to conduct the same analysis as they did for UvrD monomer to examine if there are occurrences of noninteger steps. If so, does it support the sequestering mechanism? This analysis is highly desired because it could place the sequestering mechanism in the context of a functional species that is activated.

In our second revision, we have added a new supplementary figure (Supp. Figure 8) to highlight the presence of non-integer steps in dimer unwinding. We now show example traces of fitted dimeric UvrD unwinding steps at 1 μ M ATP containing both non-integer steps and steps significantly smaller than the 3 bp average. Unfortunately, we do not have as much high quality data at these conditions as we did for monomeric UvrD at 0.5 μ M ATP, preventing us from carrying out a complete statistical analysis using kernel density estimation. Nevertheless, the prominence of non-integer steps in individual traces supports a strand sequestration mechanism for dimeric UvrD. We briefly discuss this observation in the section on the variable step size and non-integer steps of monomeric UvrD (p. 8, paragraph 2), and in the discussion section (p. 12, paragraph 3).

At the moment, we can only speculate on the mechanism of dimer stepping, because there is no consensus mechanism of dimer activation. One potential mechanism is that the leading helicase (i.e. the one at the ss/dsDNA junction) is involved in base pair unwinding and strand sequestration, but many others are possible. Extensive study will be necessary to define this mechanism precisely, which we feel is well beyond the scope of the current manuscript.

3. The sequence dependence is a contentious topic. The face value of the data in Supp. Fig. 5 is no apparent dependence on local sequence barriers, however, the interpretation of the data needs more caution. If the step size of the helicase is simply 1 bp and no other complex mechanism(s) is involved to yield an apparent bigger step size, I agree that the unwinding activity shows sequence independence. However, since the step size shows a peak at 3 bp, and in order to detect the sequence dependence of the unwinding activity, the local sequence barrier should be 3-bp or bigger. It is not fair to draw conclusions based on the presence of these local barriers that may not be sufficient at all to see the dependence of unwinding dynamics on sequence. Instead, barriers of different length should be designed. An alternative way to examine this is to study unwinding of AT base pairs and compare that with unwinding of GC base pairs, which is a practice by many biochemists in this field. The comparison at single-molecule level will be certainly more revealing: how the dwell may change with sequence and how the apparent step size may also change with the sequence et al.

We thank the reviewer for raising this point. Before we address this issue, we would first like to emphasize that the question of sequence dependence of UvrD unwinding dynamics is not a significant aspect of the manuscript, as it does not play a major role in any of our mechanistic conclusions. In the text, we mainly use the lack of correlation of step size or dwell time with hairpin position to pool the data across all positions. This procedure provides us the necessary statistics to analyze the step size and dwell time kinetics comprehensively.

Nevertheless, the reviewer raises a good point that the strand sequestration mechanism of stepping complicates analysis of sequence dependence. In thinking about this issue, we believe there are two complicating factors. First, the dwells between steps contain contributions from multiple kinetic events—ATP binding, unwinding, loop release—accumulated over several base pairs. We expect that the individual ATP-catalyzed 1-bp unwinding steps would represent the main source of DNA sequence dependence, and that any such dependence would be summed up over all base pairs that were unwound during the dwell, on average 3. Thus, as the reviewer correctly states, the local sequence variations should be >3 bp to observe sequence effects. In our non-uniform sequence, we observe both regions that act as barriers to unwinding (i.e. GC-rich regions) and that do the opposite (i.e. AT-rich regions) that are 3 bp or longer. Occurrences of GC- and AT-rich patches ≥ 3 bp can be identified in the hairpin sequence (see Supp. Table 1). In the plot of P_{open} vs hairpin position for the non-uniform sequence (Supp. Figure 2C), examples of such patches occur at the minima at 13 and 47 bp and maxima at 5 and 21 bp.

However, a second and more important complication is that, according to our looping model, our assay does not determine the position of the ss-dsDNA junction on the hairpin, but rather the last contact point of the sequestered loops to the protein (See Fig. 5). For this reason, we now realize that it would be difficult to correlate UvrD dynamics with local base-pair stability, since the dwell time at one hairpin position could depend on the stability of the ss-ds junction an undetermined number of base pairs upstream along the hairpin. As a result, we have removed our analysis of dwell time vs. P_{open} in our revision (see Supp. Figure 5), and we have tempered our statements on sequence dependence. We now state that dwell times do not correlate with hairpin position, which justifies pooling the data (p. 11, paragraph 1). We have kept our analysis of step size vs base pair stability, because it allows us to rule out “simultaneous melting” models of unwinding, in which the non-unitary step size is due to spontaneous opening of several base pairs, as elaborated in our revision (p. 7, paragraph 1). Since those models do not invoke DNA looping, the comparison of step size to P_{open} is appropriate (which is done through Supp. Fig. 5A and Supp. Fig. 2C).

An analysis of the sequence dependence of stepping dynamics, while potentially informative, would require substantial effort on our part due to these complicating factors. Given the tangential nature of

sequence dependence and the significant additional effort exploring this topic would entail, we believe these experiments are beyond the scope of the current manuscript and best reserved for future studies.

4. Lastly, regarding the discrepancy on step size measured for UvrD dimer, the question is that whether 3-bp on average is different from 4-5 bp measured using pre-steady state method. Instead of stating 4-5 bp result 'subject to large uncertainties'. It would be better to refit the published ensemble data using a kinetic mechanism that incorporates what's known, for example, to constrain the step size to 3 bases, but add additional steps that are known to occur. One obligate event in quenched flow measurements is strand dissociation, because only complete unwinding is detected and in reality there is a final step of strand dissociation. This was not included in the original analysis but now can be incorporated and determine whether the original ensemble data may also be consistent with a 3-bp mechanism. To put it in a simple way, given so much more is known about UvrD dimer now than back in 1997, can one construct an alternative kinetic mechanism that can adequately describe the data, incorporating all the knowns with a step size of 3 bases?

We maintain that these indirect estimates of step size are subject to large uncertainties. While the original 1997 pre-steady state studies estimated an unwinding kinetic step size of 4-5 bp for dimeric UvrD (see Ref. 6; referenced as 4 ± 1 bp in subsequent papers), we point out that more recent FRET measurements extract a value of 3.3 bp based on dwell time analysis of dimer unwinding of 18-bp DNA (see Ref. 25). In addition, single-molecule and ensemble fluorescence experiments have determined a kinetic step size ranging from 3.1 to 4.4 bp for monomeric UvrD activated by partner protein MutL (see Ref. 29), depending on ATP concentration. In contrast to our current measurements, the above values are all indirect estimates of step size derived from kinetics. We feel they are within a ~1-bp range and are indistinguishable from each other and our average value of 3 ± 1 bp (mean \pm std).

Considering the points discussed above, we feel it is not necessary to refit all previously published kinetic data for dimeric UvrD unwinding. In our revised manuscript, we now discuss these additional measurements in the introduction where previous estimates of the kinetic step size are discussed (p. 3, paragraph 4) and in the results section in which we compare our results to those prior estimates (p. 8, paragraph 1).

REVIEWERS' COMMENTS

Reviewer #2 (Remarks to the Author):

I appreciate the careful and thoughtful responses that the authors provided to my questions concerning the model for the rewinding phase of the helicase activity. The authors have addressed my concerns in the revised manuscript and the detailed responses to my questions. The model is more clearly described and the additional evidence supporting it strengthens the case for the model. This is interesting and compelling work and I fully support its publication.

Reviewer #3 (Remarks to the Author):

The revision addressed my major concern properly and I recommend publication.

Reviewer #4 (Remarks to the Author):

The latest revisions further support my prior review. The authors have gone to great lengths to address the comments from the other reviewers. The new revisions/explanations support the conclusions regarding UvrD step-size.